# Bridging the Gap Between Target Networks and Functional Regularization

**Alexandre Piché** [*]                                    *alexandrelpiche@gmail.com*
*ServiceNow Research*
*Mila, Université de Montréal*

**Valentin Thomas** [*]                                    *vltn.thomas@gmail.com*
*Mila, Université de Montréal*

**Rafael Pardinas**                                        *rafael.pardinas@servicenow.com*
*ServiceNow Research*

**Joseph Marino**                                          *jlouismarino@gmail.com*
*DeepMind, London*

**Gian Maria Marconi**                                     *gmmarconi@gmail.com*
*RIKEN Center for Advanced Intelligence Project*

**Christopher Pal**                                        *christopher.pal@polymtl.ca*
*Mila, Polytechnique Montréal*
*Canada CIFAR AI Chair*

**Mohammad Emtiyaz Khan**                                  *emtiyaz.khan@riken.jp*
*RIKEN Center for Advanced Intelligence Project*

**Reviewed on OpenReview:** *https://openreview.net/forum?id=BFvoemrmqX*

## Abstract

Bootstrapping is behind much of the successes of deep Reinforcement Learning. However, learning the value function via bootstrapping often leads to unstable training due to fast-changing target values. Target Networks are employed to stabilize training by using an additional set of lagging parameters to estimate the target values. Despite the popularity of Target Networks, their effect on the optimization is still misunderstood. In this work, we show that they act as an implicit regularizer which can be beneficial in some cases, but also have disadvantages such as being inflexible and can result in instabilities, even when vanilla TD(0) converges. To overcome these issues, we propose an explicit Functional Regularization alternative that is flexible and a convex regularizer in function space and we theoretically study its convergence. We conduct an experimental study across a range of environments, discount factors, and off-policiness data collections to investigate the effectiveness of the regularization induced by Target Networks and Functional Regularization in terms of performance, accuracy, and stability. Our findings emphasize that Functional Regularization can be used as a drop-in replacement for Target Networks and result in performance improvement. Furthermore, adjusting both the regularization weight and the network update period in Functional Regularization can result in further performance improvements compared to solely adjusting the network update period as typically done with Target Networks. Our approach also enhances the ability to networks to recover accurate $Q$-values. The code is available here `https://github.com/AlexPiche/fr-tmlr/`.

# 1 Introduction

Value functions are at the core of deep Reinforcement Learning (RL) algorithms. In order to learn a value function via regression, we need to estimate the target value of the state. Estimating the target value using Monte Carlo roll-outs requires complete trajectories and can have high variance. These issues make value estimation via Monte Carlo inefficient and unpractical in complex environments. Temporal Difference (TD) (Sutton, 1988) algorithm addresses these issues by using *bootstrapping* (Sutton & Barto, 2018). Specifically, the value of a state is estimated using the immediate reward and the discounted predicted value of its successor. Bootstrapping has potentially lower variance than Monte Carlo, particularly in the off-policy settings, and does not require entire trajectories to estimate the target value. Bootstrapping can unfortunately be unstable as the target value is estimated with constantly updated parameters.

While stabilizing off-policy value function learning has been an active area of research (Precup, 2000; Precup et al., 2001; Sutton et al., 2008; 2009). We focus on *Target Networks* (TN) (Mnih et al., 2013) a popular technique in deep RL which uses an additional set of lagging parameters to estimate the target value. Today TNs are central to most modern deep RL algorithms (Mnih et al., 2013; Lillicrap et al., 2015; Abdolmaleki et al., 2018b; Haarnoja et al., 2018; Fujimoto et al., 2018; Hausknecht & Stone, 2015; Van Hasselt et al., 2016; Hessel et al., 2018). Their popularity is due to the ease of implementation, little computation overhead, and their demonstrated effectiveness in a range of domains. Despite their popularity, the inner workings of TNs and how they stabilize the optimization of the value function remain unclear. While Van Hasselt et al. (2018) empirically shows that TNs help but do not prevent divergence, Zhang et al. (2021) shows that a variant of TNs can prevent divergence in simple cases.

In this work, we shed light on the regularization implicitly performed by TNs and how it is related to a regularization in the $Q$-function space whose strength is dictated by the discount factor $\gamma$. However, this regularization cannot be written as the gradient of a convex regularizer, as is commonplace in optimization. We prove that, because of this, TNs can unstabilize TD instead of making it more stable. This leads us to propose a simple Functional Regularization (FR) alternative that has the advantages of provably ensuring TD remains stable while being more flexible than TN.

In our experimental study, we explored a variety of environments, including the two-state MDP (Tsitsiklis & Van Roy, 1996), the Four Rooms environment (Sutton et al., 1999), and the Atari suite (Bellemare et al., 2013), to assess the efficacy of regularization introduced by TN and FR in relation to performance, accuracy, and divergence. Our findings emphasize that Functional Regularization without regularization weight tuning can be used as a drop-in replacement for Target Networks without loss of performance and can result in performance improvement. Additionally, the combined use of the additional regularization weight and the network update period in FR can lead to enhanced performance compared to merely tuning the network update period for TN.

**Contributions:**

1. Target Network induces a pseudo-regularization which effectively can slow down the Temporal Difference (TD) optimization process. However, it can also unstabilize TD, i.e, make the iteration divergent while TD alone would converge.
   **Evidence:** Explanation in Section 3 and example in Section 4.1

2. There is a continuous spectrum between FR and TD(0) and as such, for $\kappa$ small enough FR has the same convergence properties as TD.
   **Evidence:** Proposition 3.2 and its proof in Appendix A.3.

3. FR can result in improved performance when compared to TN and TD(0) in a variety of environments, discount factors, and levels of "off-policiness".
   **Evidence:** Demonstrated empirically on the Four Room environment (Section 4.2.2) and the Atari suite (Section 4.3, Section 4.4.2).

## 2  Background

**Preliminaries.** We consider the general case of a Markov Decision Process (MDP) (Puterman, 2014) defined by $\{\mathcal{S}, \mathcal{A}, p, r, \gamma, \mu\}$, where $\mathcal{S}$ and $\mathcal{A}$ respectively denote the finite state and action spaces, $p(\mathbf{s}'|\mathbf{s}, \mathbf{a})$ represents the environment transition dynamics, i.e the distribution of the next state taking action $\mathbf{a}$ in state $\mathbf{s}$. $r: \mathcal{S} \times \mathcal{A} \to \mathbb{R}$ denotes the reward function, $\gamma \in [0, 1)$ is the discount factor, and $\mu$ is the initial state distribution. For a given policy, $\pi: \mathcal{S} \to \Delta(\mathcal{A})$, a starting state $\mathbf{s}$ and action $\mathbf{a}$, the value function $Q^\pi \in \mathbb{R}^{|\mathcal{S}| \times |\mathcal{A}|}$ is the expected discounted sum of rewards:

$$Q^\pi(\mathbf{s}, \mathbf{a}) = \mathbb{E}_\pi \left[ \sum_{t \geq 0} \gamma^t r(\mathbf{s}_t, \mathbf{a}_t) | \mathbf{s}_0 = \mathbf{s}, \mathbf{a}_0 = \mathbf{a} \right]. \tag{1}$$

Additionally, the value function satisfies the Bellman equation (Bellman, 1966)

$$Q^\pi = R + \gamma P^\pi Q^\pi \equiv \mathcal{T} Q^\pi \tag{2}$$

where $R \in \mathbb{R}^{|S| \cdot |A|}$ the reward vector and the state-action to state-action transition matrix $P^\pi \in \mathbb{R}^{|\mathcal{S}| \cdot |\mathcal{A}| \times |\mathcal{S}| \cdot |\mathcal{A}|}$ where $P^\pi \big[(\mathbf{s}, \mathbf{a}), (\mathbf{s}', \mathbf{a}')\big] = \pi(\mathbf{a}'|\mathbf{s}') \cdot p(\mathbf{s}'|\mathbf{s}, \mathbf{a})$ and $\mathcal{T}: \mathbb{R}^{|\mathcal{S}| \cdot |\mathcal{A}| \times |\mathcal{S}| \cdot |\mathcal{A}|} \to \mathbb{R}^{|\mathcal{S}| \cdot |\mathcal{A}| \times |\mathcal{S}| \cdot |\mathcal{A}|}$ is the Bellman operator.

**Linear Function Approximation (LFA).** The value function $Q^\pi$ can be approximated via a linear function. For a fixed feature matrix $\Phi \in \mathbb{R}^{|S| \cdot |A| \times p}$, our aim is to learn a parameter vector of dimension $p$, $w \in \mathbb{R}^p$, so that $Q_w \equiv \Phi w$ approximates $Q^\pi$. We denote the off-policy sampling distribution by the diagonal matrix $D \in \mathbb{R}^{|S| \cdot |A| \times |S| \cdot |A|}$ with diagonal entries $d(s, a)$, positive and summing to 1. The update rule given by the TD(0) semi-gradient (Sutton & Barto, 2018) is given by

$$\tilde{\nabla}_w \ell^{\mathrm{TD}}(w) = -\Phi^\top D(R + \gamma P^\pi \Phi w - \Phi w) \tag{3}$$

where the remainder of the return is estimated via bootstrapping, i.e., $P^\pi \Phi w$.

**Deep Neural Networks (DNNs) Approximation.** In complex environments, it is difficult to adequately design features to approximate $Q^\pi$ with a linear function. Instead, non-linear functions, such as DNNs, are used to parametrize $Q_w$ to automatically learn features and to improve the approximation to $Q^\pi$. But using DNNs to estimate $Q^\pi$ using off-policy data and bootstrapping can be unstable and diverge (Sutton & Barto, 2018; Van Hasselt et al., 2018). It is common to use a lagging set of weights $\bar{w}$ to estimate the next value function to mitigate divergence (Mnih et al., 2013; Lillicrap et al., 2015). The network parametrized by the set of lagging weights is commonly referred to as a target network. Despite the popularity of target networks, there are still gaps in our understanding of the impact it has on learning dynamics.

## 3  Functional Regularization as an Alternative to Target Networks

### 3.1  Understanding Target Network's implicit regularization

In this subsection, we will look more closely at the effects TNs have on the optimization. We place ourselves in the classical Linear Function Approximation (LFA) setting (Sutton & Barto, 2018) in order to be able to make theoretical contributions.

*Target Networks* (Mnih et al., 2013) are a popular approach to stabilize Temporal Difference learning. A periodically updated copy, $Q_{\bar{w}}$, of $Q_w$ is used to estimate the next state value. This causes the regression targets to no longer directly depend on the most recent estimate of $w$. With this, the Mean Squared Bellman Error for a given state transition is

$$\ell^{\mathrm{TN}} = \tfrac{1}{2} \| R + \gamma P^\pi Q_{\bar{w}} - Q_w \|_D^2. \tag{4}$$

The semi-gradient (Sutton & Barto, 2018), denoted by $\tilde{\nabla}$, of the expected squared Bellman error with a TN can be decomposed as (see proof in Appendix A)

$$\tilde{\nabla}_w \ell^{\mathrm{TN}}(w) = \underbrace{-\Phi^\top D(R + \gamma P^\pi \Phi w - \Phi w)}_{\mathrm{TD}(0)} + \underbrace{\gamma\ \Phi^\top D P^\pi \Phi(w - \bar{w})}_{\text{``Regularizer''}}. \tag{5}$$

Written in this form, the effects of using TNs become clearer. The first term is the usual TD(0) update that we will not attempt to modify in this paper, and the second term can be interpreted as a "regularization" term that encourages the weights $w$ to stay close to the frozen weights $\bar{w}$. This is in line with the intuition that TNs "slow down" or "stabilize" the optimization. However, this "regularizer" suffers from two main issues.

- The first one is the weighting of the regularization which is equal to the discount factor $\gamma$. This may be problematic as $\gamma$ controls the agent's effective horizon (Sutton & Barto, 2018) and is part of the problem definition. Therefore, TNs are inflexible in the sense that the weight of the regularizer cannot be controlled independently from the agent's effective horizon, and can only be controlled through the target update period $T$. We show experimentally in Section 4.2 that the update period $T$ can be ineffective at controlling the strength of the regularization.

- The second and most important point is that the "regularization" is not a proper regularization term. It may be interpreted as the gradient of a quadratic when $\Phi^\top D P^\pi \Phi$ is symmetric and positive, which is not the case in general. The underlying issue is that if $\Phi^\top D P^\pi \Phi$ has a negative eigenvalue, the "regularizer" will actually push $w$ away from $\bar{w}$ instead of bringing them closer and thus will render the whole optimization unstable. **This is an important theoretical issue as usually a basic requirement for a regularizer is that it cannot cause the original method to diverge.** It is verified in Section 4.1 that TN can create additional divergence zones that were not present in TD(0).

Both aforementioned issues can be resolved simply. For the first point, we replace $\gamma$ with a separate hyperparameter $\kappa \geq 0$. As for the second point, changing $\Phi^\top D P^\pi \Phi$ to $\Phi^\top D \Phi$ ensures that the second term is a proper quadratic regularizer that cannot destabilize the optimization (Tihonov, 1963; Tikhonov, 1943; Lyapunov, 1992).

## 3.2 Introducing Functional Regularization

Following the reasoning developed in the previous subsection, we would like to design a loss $\ell$ whose semi-gradient is

$$\tilde{\nabla}_w \ell(w) = \Phi^\top D (R + \gamma P^\pi \Phi w - \Phi w) + \kappa \ \Phi^\top D \Phi (w - \bar{w}). \tag{6}$$

The term $\kappa \ \Phi^\top D \Phi (w - \bar{w})$ is the gradient of the quadratic form

$$\frac{\kappa}{2}(w - \bar{w})^\top \Phi^\top D \Phi (w - \bar{w}) = \frac{\kappa}{2}\|Q_w - Q_{\bar{w}}\|_D^2$$

as $Q_w = \Phi w$. We see that the natural resulting regularizer is a Functional Regularizer in norm $\|\cdot\|_D$, the usual norm when studying the convergence of TD in RL (Bertsekas & Tsitsiklis, 1996). This leads to the *Functionally regularized* Mean Square Bellman Error which penalizes the norm between the current Q-value estimate $Q_w(\mathbf{s}, \mathbf{a})$ and a lagging version of it $Q_{\bar{w}}(\mathbf{s}, \mathbf{a})$, giving us

$$\ell^{\mathrm{FR}}(w) = \frac{1}{2}\|R + \gamma \lceil P^\pi Q_w \rceil - Q_w\|_D^2 + \frac{\kappa}{2}\|Q_w - Q_{\bar{w}}\|_D^2, \tag{7}$$

where $\lceil \rceil$ denotes the stop-gradient operator. We observe that the up-to-date parameter $w$ is used in the Squared Bellman Error and there is the additional decoupled FR loss to stabilize $w$ by regularizing the $Q$ function. $\kappa \geq 0$ is the regularization parameter and we recover TD learning for $\kappa = 0$.

Critically, unlike Equation (4), the target $Q$-value estimates are supplied by the up-to-date $Q$-network, with the explicit regularization now separately serving to stabilize training. As with a TN, we can update the lagging network periodically to control the stability of the $Q$-value estimates. Overall, $\ell^{\mathrm{FR}}$ is arguably similar in complexity to $\ell^{\mathrm{TN}}$, requiring only an additional function evaluation for $Q_w(\mathbf{s}_{t+1}, \mathbf{a}_{t+1})$ and an additional hyper-parameter, $\kappa$. We report an execution time analysis of replacing TN with FR in Figure 6. While the gradient update is now slightly longer on average, in a training scenario a large portion of the time is spent on inference and simulation which make the difference in gradient update time negligible.

### 3.3 Stability of Target Network regularization and Functional Regularization

Now that we have highlighted potential issues that could arise when using TNs and proposed an alternative objective function, we will show that indeed TN can *unstabilize* Temporal Difference learning while FR, under some conditions, does not.

**Theorem 3.1** (Target Networks can **unstabilize** TD)**.** *Using TD(0) with Target Networks ( Equation* (4)*) can diverge while simply using TD(0) would converge.*

A simple example of this situation is exhibited in Section 4.1. Note that in Appendix A.2 we precisely characterize the domain of convergence of TD(0) with TN for large $T$ and show that it changes the convergence domain of TD(0), leading to the theorem above.

On the other hand, for FR, we have

**Proposition 3.2** (Convergence of Temporal Difference with Functional Regularization)**.** *If $\Phi, D, P^\pi$ are chosen so that TD(0) converges, then for $\kappa$ small enough, for $T$ large enough, TD(0) with Functional Regularization is guaranteed to converge.*

*Proof.* The proof can be found in Appendix A.3. $\square$

The results above highlight the intuition of the last subsections: while TNs can slow down learning they also interfere with TD learning, changing its convergence domain. On the other hand, FR with appropriately chosen parameters does not impact convergence while still being able to regularize the algorithm.

Finally, note here that **results above do not mean that using TD(0) with FR is always more stable than using it with TN**. When TD(0) is already unstable, depending on the interplay between the features, the off-policy distribution, and the transition matrix, it would be possible in some cases for TNs to make the iteration stable. However, this is unlikely and shows that it is impossible when the features are low dimensional (see Corollary A.4).

**Connection to GTD methods:** Similarly to TN and FR updates Equation (5) and Equation (6), gradient TD methods (Antos et al., 2008) such as TDC and GTD2 (Sutton et al., 2009) make use of a second set of weights. This second set of weights is used to add a correction term to the TD update so that it approximates a gradient update, thus not suffering from the divergence induced by the deadly triad. The correction term of gradient TD methods has strong similarities with the TN correction $\gamma\Phi^\top DP\Phi(w - \bar{w})$ of Equation (5). However the additional set of weights used in gradient TD methods is the result of a least squares optimization process and approximates a projection of the TD error for the current state. Therefore, the theoretical convergence of these methods rely on a two-timescale argument where the second set of weights must converge close to their optimal values. In practice, it seems that these algorithms work best when the learning rate for the second set of weights is very small (Ghiassian et al., 2020), thus making these gradient TD methods behave like TD(0). Furthermore, the update for the second set of weight needs importance sampling in the off-policy case and adapting the algorithm to deep learning is not straightforward (Silver, 2013). Thus these methods, while enjoying some theoretical guarantees, are not practical for most modern settings and are quite different from FR. FR and TN, by working directly in the function space, avoid complications related to being used with deep neural networks. However neither FR nor TN enjoy strong convergence guarantees in presence of the deadly triad as we have shown above.

### 3.4 From value estimation to control

While in the previous subsections we have discussed the value estimation case, we can also use FR for control, for instance with Q-learning. As the update in Q-learning can be seen as a TD update where the next $Q$-value is sampled from the greedy policy, FR can straightforwardly be adapted for control. We illustrate the practical difference between using FR or TN for control in Algorithm 1.

In Algorithm 1 we provide code for a simple Q-learning agent using either Target Networks or Functional Regularization. Note that the change is minimal and adds little complexity. For our experiments in Section 4.2.2,

---

**Algorithm 1** Deep Q-Network (DQN) Algorithm with TN or FR

---

1: Initialize replay memory $\mathcal{D}$ with capacity $D$
2: Initialize action-value function $Q$ with random weights $\theta$
3: Initialize target weights $\bar{\theta} \leftarrow \theta$
4: **for** episode = 1 to $M$ **do**
5:     Initialize state $s \sim \mu$
6:     **for** t = 1 to $N$ **do**
7:         With probability $\epsilon$ select a random action $a$, otherwise $a \leftarrow \text{argmax}_a Q_\theta(s, a)$
8:         Execute action $a$ in environment, observe reward $r$ and next state $s'$
9:         Store transition $(s, a, r, s')$ in $\mathcal{D}$
10:        Sample random mini-batch of transitions $(s_j, a_j, r_j, s'_j)$ from $\mathcal{D}$
11:        *// Set the target*
12:        TN: $y_j \leftarrow r_j + \gamma \max_{a'} Q_{\bar{\theta}}(s'_j, a')$
13:        FR: $y_j \leftarrow r_j + \gamma \max_{a'} Q_\theta(s'_j, a')$
14:        **If** $s'_j$ is terminal **then** $y_j \leftarrow r_j$
15:        *// Define the loss*
16:        TN: $\ell(\theta) = \frac{1}{2}\left(y_j - Q_\theta(s_j, a_j)\right)^2$
17:        FR: $\ell(\theta) = \frac{1}{2}\left(\lceil y_j \rceil - Q_\theta(s_j, a_j)\right)^2 + \frac{\kappa}{2}\left(Q_\theta(s_j, a_j) - Q_{\bar{\theta}}(s_j, a_j)\right)^2$
18:        Perform a gradient descent step on $\ell(\theta)$ with respect to $\theta$
19:        Every $T$ steps set $\bar{\theta} \leftarrow \theta$
20:        *// Update state*
21:        **If** $s'$ is terminal **then** $s \sim \mu$, **else** $s \leftarrow s'$
22:     **end for**
23: **end for**

---

Section 4.3 and Section 4.4 we use a DQN variation (Mnih et al., 2013) of this algorithm which include sampling mini-batches from a replay buffer, using the Adam optimizer (Kingma & Ba, 2014) and using an $\epsilon$-decay.

## 4 Experiments

In this section, we investigate empirically the differences between FR, TN and TD(0). First, we aim to understand the convergent and divergent behaviors of these algorithms. We use a novel visualization to give intuition and validate our theoretical results. Finally, we investigate if the use of the additional regularization weight $\kappa$ combined with the network update period $T$ can lead to better performance compared to vanilla Deep Q-learning (DQL) and to merely tuning the network update period $T$ for TN for varying degrees of off-policiness and discount factor on the Four Room environment (Sutton, 1988) and the Atari benchmark (Bellemare et al., 2013).

### 4.1 Simple 2-state MDP - Off Policy Evaluation

### 4.1.1 Experimental Set-Up

In this subsection, we visualize the behavior of TN and FR on a simple MDP. This illustrates the theory developed in Section 3 on the convergence properties of both algorithms. Baird (1995); Tsitsiklis & Van Roy (1996) and Kolter (2011) introduced simple reward-less MDPs with few states on which TD(0) can exhibit divergence. Similarly, we define a 2-state MDP in Figure 1a, which is conceptually close to Tsitsiklis & Van Roy (1996). Each state transitions to the other with probability 0.95 or transitions to itself with probability 0.05. We use 1-d features for these two states: $\phi(s_0)$ and $\phi(s_1)$. We denote the off-sampling distribution of $s_0$ by $d(s_0)$ and by $\pi(s_0)$ the on-policy sampling distribution of $s_0$ which is equal to 1/2 in this specific case. We use $\gamma = 0.995$.

Instead of just exhibiting a single instance $D, \Phi$ for which we have divergence, we showcase the convergent or divergent behavior of TD(0), TD(0) with TN and TD(0) with FR *for all combinations* of $D$ and $\Phi$

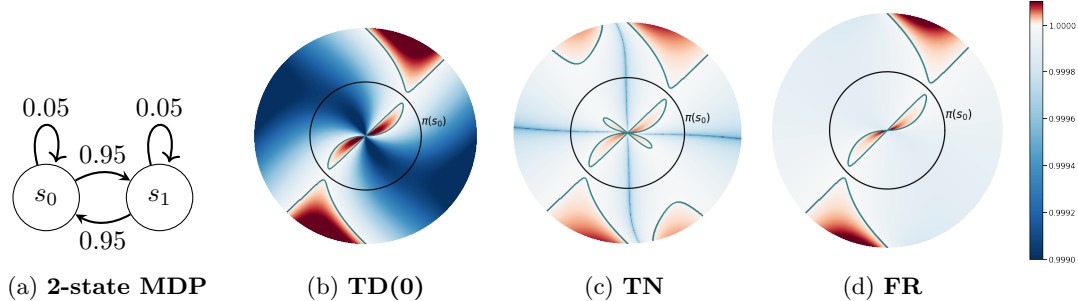

(a) **2-state MDP**    (b) **TD(0)**    (c) **TN**    (d) **FR**

Figure 1: **Convergence rate for different off-policy and feature combination in a 2-state MDP.** We show here the spectral radius for TD(0) (b) only, (c) with Target Network, and (d) with Functional Regularization on the simple MDP (a). For any point in the disk, its radius represents the off-policy distribution of $s_0$ while the angle is the same as the vector $[\phi(s_0), \phi(s_1)]$. The color represents the value of the spectral radius for each method while the contour is the frontier $\rho = 1$ separating the convergence domain (shades of blue) from the divergent domain (shades of red). The black circle has a radius $\pi(s_0)$, the stationary distribution of $P$ for which TD is guaranteed to converge as it would be on-policy. The fact that Target Networks might cause policy evaluation to diverge for some features where simple TD converges is a surprising and important limitation of Target Networks given their ubiquity.

simultaneously. The Euclidean norm of $\Phi$, $\|\Phi\|_2$ does not impact the convergence of any of the aforementioned algorithms, as $\Phi$ can be renormalized if we scale the learning rate $\eta$ by its squared norm. Therefore, only the angle of the vector $\Phi = [\phi(s_0), \phi(s_1)]$ is needed to describe all representations. In our context, there are only two states, and thus the off-policy distribution is entirely determined by $d(s_0)$. Thus, one can represent the set of all representations and all off-policy distributions $(D, \Phi)$ as a disk of radius 1 [1]. In polar coordinates, the radius of a point on the disk represents $d(s_0)$ while its angle is the one of the vector $\Phi$, i.e., $\varphi(\Phi) = \arctan \frac{\phi(s_1)}{\phi(s_0)}$.

### 4.1.2   Results

We draw the above-mentioned disks for the algorithms: TD(0), TD(0) with TN, and TD(0) with FR (Figure 1b, 1c, and 1d). On these disks, the colors represent the spectral radius $\rho$ of the iteration matrix for each algorithm. Recall that $\rho < 1$ (blue) implies convergence, while $\rho > 1$ (red) implies divergence to $+\infty$. We plot the contour lines $\rho = 1$ to facilitate visualization. Additionally, we draw the circle of radius $d(s_0) = \pi(s_0)$ (in black), which denotes the stationary distribution. As predicted by theory, TD(0) converges on this circle for all representations. We use the regularization coefficient $\kappa = 1.5$ and target update period $T = 10,000$ for TD(0) with TN and TD(0) with FR.

As seen in Figure 1, FR (Figure 1d) and TN (Figure 1c) show less extreme values (lighter blue) compared to TD(0) (Figure 1b). This finding implies that both convergence and divergence are slowed down, which is expected for regularization methods.

We recover the classical divergence example for TD(0) (Tsitsiklis & Van Roy, 1996; Van Hasselt et al., 2018): if $\phi(s_1) > \phi(s_0) > 0$ and the behavior distribution $d(s_0)$ is close to 1, the point $(d(s_0), \varphi(\Phi))$ in polar coordinates lies in the large upper red region of Figure 1b, thus TD(0) will diverge.

As proved in Proposition 3.2, the convergence domain of TD(0) with FR is similar to the one of TD(0). Interestingly, and as shown in Theorem 3.1, TD(0) with TN may diverge in regions where TD(0) with FR does not. For example, in Figure 1c we see four smaller (symmetric) additional divergence regions (red) in the lower right and upper left regions of the disk for $d(s_0)$ close to 1 and two symmetric ones near the center of the disk. This can happen when $\Phi^\top D P^\pi \Phi$ which appears in the implicit regularization of TN has negative eigenvalues.

**Impact of "Off-Policyness":** As previously reported by Kolter (2011) and Scherrer (2010), we also observe that when the data collection is on-policy (black circle of radius $\frac{1}{2}$) for every method and every

---

[1]Note that we technically only need a half-disk to illustrate the convergence behavior, but we plot the whole disk for aesthetic reasons.

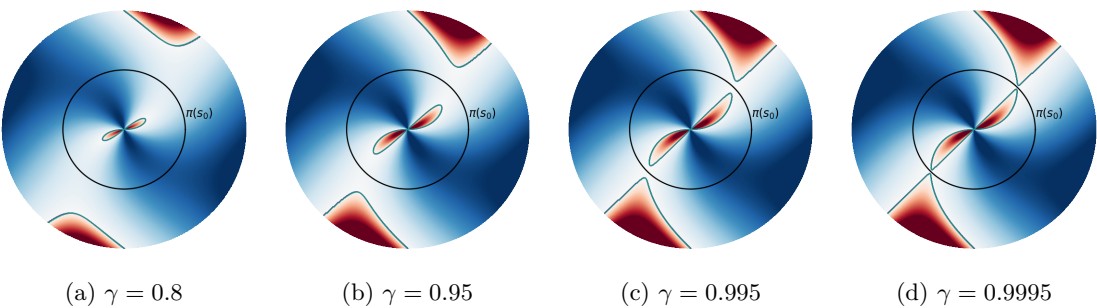

$$(a) \ \gamma = 0.8 \qquad (b) \ \gamma = 0.95 \qquad (c) \ \gamma = 0.995 \qquad (d) \ \gamma = 0.9995$$

Figure 2: **Influence of the discount factor $\gamma$ on TD(0) divergence regions.** As we increase $\gamma$, the area of the divergence regions (in red) increases. Specifically, as $\gamma$ approaches 1, the divergence regions almost extend to the on-policy case, meaning that some behavior policies close to the on-policy distribution that might have been stable for lower $\gamma$ become unstable for higher $\gamma$. A similar behavior is also observed for TN and FR.

discount factor $\gamma$ the method does not diverge. However as we move away from the on policy stationary distribution (increasing or decreasing the radius) the methods have divergence zones.

**Impact of the discount factor $\gamma$:** It is known that $\gamma$ plays a strong role in the divergence of TD methods (Kolter, 2011; Scherrer, 2010). In Figure 2 we provide an exact illustration of this relationship between the discount factor $\gamma$ and the convergence or divergence behavior of Temporal Difference methods in our simple MDP. Intuitively, as the divergence issues stem from the fact that $\gamma P^\pi$ is non-symmetric, we can expect $\gamma \to 0$ to be stable as our problem would simplify into a weighted least squares problem. On the other hand, the divergence issues could be made worse for higher $\gamma$. This intuition is validated on Figure 2 where we can see that the divergence regions (in red) increase with $\gamma$ for TD(0). We provide similar figures for FR and TN in Appendix C.2. Interestingly, as on-policy TD(0) is always stable, we observe that for smaller $\gamma$ we can afford to be more off-policy than for higher $\gamma$ where any small deviation from the on-policy distribution could lead to divergent behavior, depending on the features $\phi$.

These depictions of the convergence or divergence behaviors of TD(0) with FR and TD(0) with TN hint at the fact that TN-based algorithms may be more unstable than FR-based algorithms for strongly off-policy distributions. This hypothesis is studied empirically in the next subsection.

## 4.2 Four Rooms Results

### 4.2.1 Experimental Set-Up

In this section, we study the behavior of vanilla DQL, TN, and FR in the Four Room environment (Sutton et al., 1999). The environment is comprised of four rooms divided by walls with a single gap in each wall allowing an agent to move between rooms. The environment size is $11 \times 11$. The agent's position is given by a one-hot encoding, and its available actions are up, down, left, and right. The agent starts in the bottom right position and must reach the goal in the top left position to obtain a reward of 1. At each step that the agent does not reach the goal, it receives a small penalty of 0.01. The episode terminates after 121 ($=11^2$) steps, thus an agent that never reach the goal would have a return of $-1.21$.

We investigate how varying the discount factor $\gamma$ and exploration level $\epsilon$ can lead to divergence in both their $Q$-value error and results. Specifically, the data is collected under a behavior policy that takes a random action with probability $\epsilon$ and takes $\mathbf{a} = \arg\max_{\mathbf{a}} Q_w(\mathbf{s}, \mathbf{a})$ with probability $1 - \epsilon$. As $\epsilon$ tends towards 1, the data collection becomes more off-policy and makes the $Q$-value estimation more difficult. Similarly, as shown above, increasing the discount factor $\gamma$ increases the difficulty of the bootstrapping and can lead to divergence.

We collect 10000 environment transitions and perform the same number of gradient steps. At the end of training, we report the average per-episode regret for 100 episodes collected under $\epsilon_{\text{eval}}-$greedy policy with $\epsilon_{\text{eval}} = 0.1$ as it is common for RL agents in deterministic environments such as the Atari suite. Furthermore, to better understand the behavior of the algorithms studied, we measure $\frac{1}{n}\|Q_w - Q^*\|_2^2$, where $Q^*$ was obtained

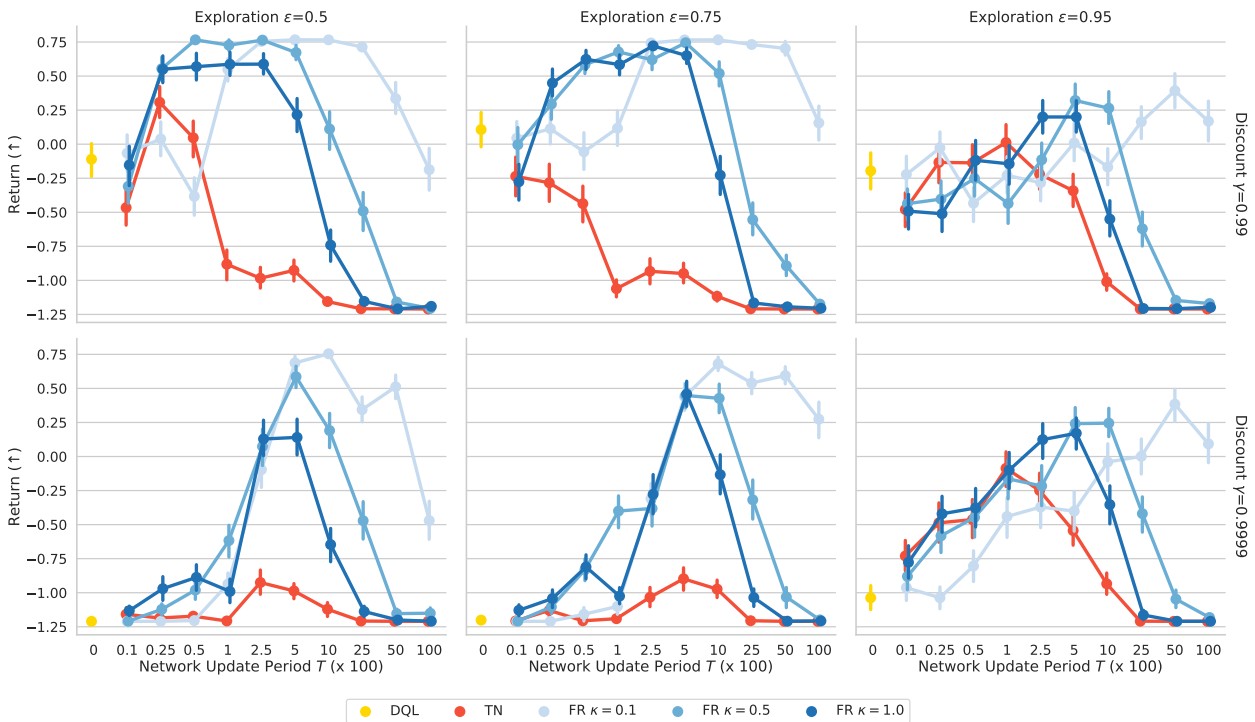

(a) **Final return for different hyper-parameters (Higher better)** We report the average performance over 100 episodes for 40 seeds on the Four Room environment. We observe that FR outperforms TN for most combination of update period $T$ and regularization weight $\kappa$ across every scenario studied.

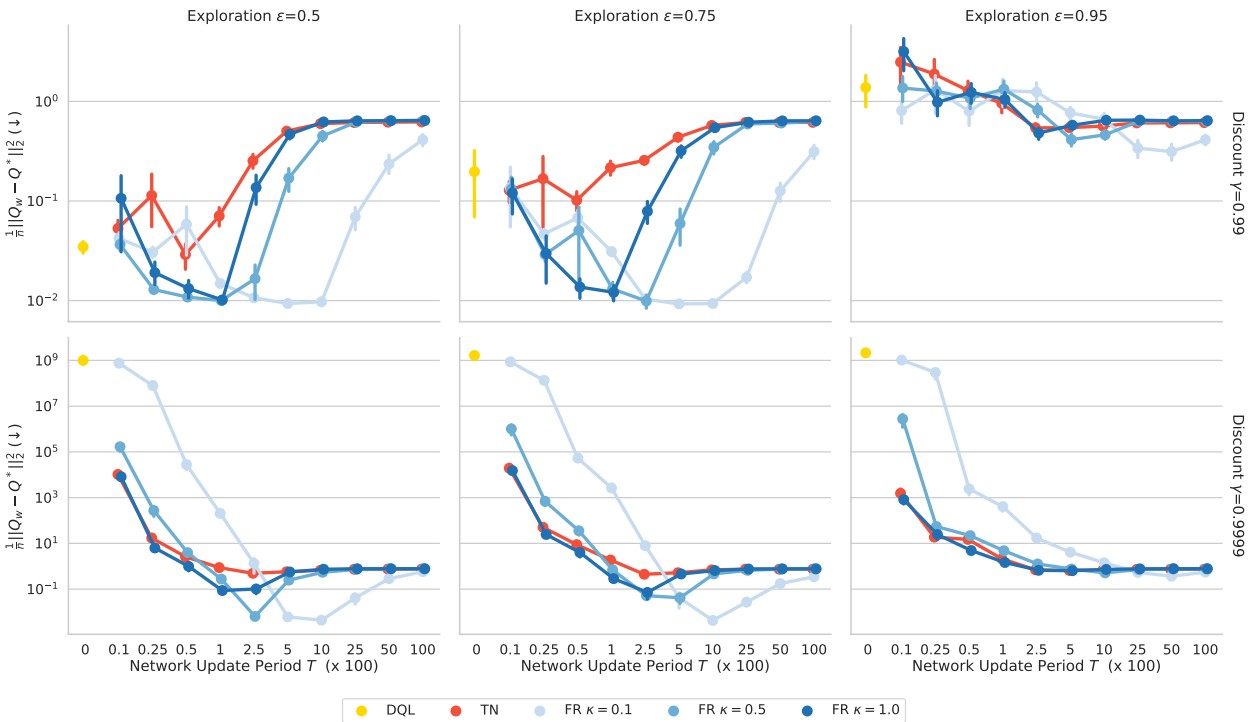

(b) **Final loss for different hyper-parameters (Lower better)** We report the difference between the $Q$-value obtained using function approximation $Q_\theta$ and the optimal $Q$-value $Q^*$ estimated using a tabular method. We can see that as the error decreases the performance increases (Figure 3a). We also observe that FR $\kappa = 0.1$ obtains lower $Q$ errors than TN for every scenario studied.

Figure 3: Four Room Ablation Study

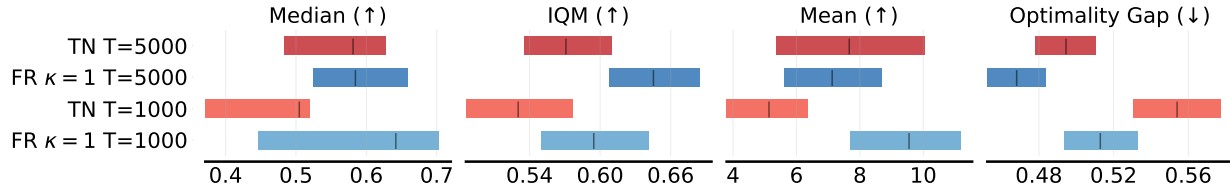

Figure 4: **Atari Suite Human Normalized Score.** Different metrics for the Human Normalized Score with their 90% confidence interval on the whole Atari suite.

using tabular $Q$-learning, $Q_w$ is a DNN approximation (details in Table 1), and $n$ is the number of state-action pairs. We repeat the evaluation for 40 seeds. We investigate the performance of vanilla DQL, TN and FR under 6 different combinations of discount factor $\gamma \in \{0.99, 0.9999\}$ and exploration $\epsilon \in \{0.5, 0.75, 0.95\}$.

### 4.2.2 Results

The returns and the $Q$-value errors for the Four Room environment can be observed in Figure 3a and Figure 3b respectively.

**Increasing exploration $\epsilon$.** Increasing exploration $\epsilon$ from 0.5 to 0.75 does not result in loss of performance or worst $Q$-value approximation for the FR algorithms (in shades of blue). However, TN's (red) performance and its $Q$ value approximation degrade. Increasing exploration $\epsilon$ past 0.75 to 0.95 decreases FR's performance and $Q$-value approximation. Interestingly, for exploration $\epsilon = 0.95$ slightly increases TN performances, but decreases its $Q$-value approximation accuracy. The performance of vanilla DQL (yellow) stays fairly constant as we increase exploration $\epsilon$, but for discount $\gamma = 0.99$ its $Q$-value error increases, while it diverges for every exploration level for discount $\gamma = 0.9999$.

**Increasing the discount factor $\gamma$.** For smaller exploration $\epsilon$, increasing the discount factor $\gamma$ from 0.99 to 0.9999 results in much worst performance and $Q$-value divergence for small network update periods $T$. While several different combinations of network update period and regularization resulted in best performance for discount factor $\gamma = 0.99$, large network update periods and small regularization weights result in better performance and smaller $Q$-error for $\gamma = 0.9999$. The performance and $Q$-value approximation of TN and vanilla DQL degrades greatly. For exploration $\epsilon = 0.95$, increasing the discount factor also causes the $Q$-value to diverge for smaller network update periods $T$, but interestingly does not result in a large decrease in performance for larger network update period $T$ for both TN and FR, while the performance of vanilla DQL collapses.

**Overall performance.** In Figure 3a, for discount rate $\gamma = 0.99$, there are a number of regularization weights $\kappa$ and network update periods $T$ hyper-parameters for which FR reaches optimal or near optimal performance, suggesting that our method is not particularly sensitive to the choice of the hyper-parameter $\kappa$. However, **for exploration $\epsilon \leq 0.75$ and for discount rate $\gamma = 0.9999$, we observe that optimal performance and $Q$-value approximation can only be reached through a combination of regularization weight $\kappa$ smaller than 1 and large network update period $T$. Highlighting the benefits of tuning the regularization weight $\kappa$ and network update period $T$ for FR compared to simply tuning the network period update for TN.** Furthermore, we note that in every settings studied, FR outperforms both vanilla DQL and TN. Finally, for every method we observe an inverted "U" shape that highlight the regularization trade-off where too little regularization results in divergence, while too much regularization results in slow learning.

### 4.3 Atari Suite

### 4.3.1 Experimental Set-Up

In this section, we investigate if FR without additional tuning ($\kappa = 1$) can be used as a drop-in replacement for TN on the Arcade Learning Environment (ALE) (Bellemare et al., 2013). We use the `CleanRL` library (Huang

et al., 2022), and we run each algorithm for 10M steps and use 7 seeds for each game in the suite which amounts to a total of approximately $60,000$ GPU hours. We keep every hyper-parameters to their default value with the exception of the network update period $T$ that we varied.

While all learning curves for all games are available in Figure 10, we use the `rliable` library and the robust metrics introduced in (Agarwal et al., 2021) to analyze results over all games. In Figure 4, we report the usual mean and median scores, as well as the robust metrics interquartile mean (IQM) and optimality gap as well as their 90% confidence interval. The **IQM** is the mean where the top and bottom 25% of the runs are discarded. It can be understood as halfway between the mean and the median as the mean would not discard any runs and the median would discard the top and bottom 50% of the runs to only keep the median. An additional robust metric is the **optimality gap** which measures how far the algorithm is from reaching a score of 1, the average human score, on all games.

### 4.3.2 Results

**Network update period** $T = 1,000.$ We first compare FR $\kappa = 1$ to TN with the default network update period $T = 1,000$. We observe that the mean is significantly higher for FR $\kappa = 1$ (light blue) than for TN (light red). Furthermore, the average median and the IQM are higher for FR than TN, but their distribution are somewhat overlapping, we also note that the optimality gap for FR is noticeably better than the one achieved by TN.

**Network update period** $T = 5,000.$ We then compare FR $\kappa = 1$ to TN for the best network update period $T = 5,000$ obtained for TN in Section 4.4. We observe that the mean and median distributions are mostly overlapping for TN (dark red) and FR $\kappa = 1$ (dark blue). While we cannot claim statistical significance for our sample size, we note that the IQM is higher and optimality gap is lower for FR $\kappa = 1$ than TN. Furthermore, we note that despite FR $\kappa = 1$ $T = 5000$ does not significantly improve upon TN $T = 5000$ for our sample size, we observed noticeable performance improvement on the following games: atlantis, enduro, kung fu, zaxxon, crazy climbers, seaquest, frostbite, and robotank.

**Overall performance.** We conclude that for the network update periods tested, **FR matches or outperforms TN on most games of the ALE without additional hyper-parameter tuning (setting $\kappa = 1$).** Further performance improvement could potentially be reached by tuning $\kappa$ in concert with the target network update period $T$ as done in the Four Room environment.

### 4.4 Sensitivity Analysis of the Regularization Parameters on a Subset of Atari Games

#### 4.4.1 Experimental Set-Up

In this section, we investigate if the findings from the previous sections hold for diverse and challenging environments from the Arcade Learning Environment (Bellemare et al., 2013). Specifically, we investigate the behavior of deep $Q$-learning under the following 6 environments from the orignal DQN paper (Mnih et al., 2013): Seaquest, Breakout, Space Invaders, Enduro, Qbert and Beam Rider[2]. For each environment, we decay the probability of a random action from 1 to $\epsilon$ and the discount factor $\gamma$ use to train the Q-value. We report the results for different $\gamma$ and $\epsilon$ since they can both increase instability and result in divergence. Unless specified otherwise, we use the default hyper-parameters from the CleanRL library (Huang et al., 2022).

We study the average return of the $\epsilon_{\text{eval}}$-greedy policy after 10M environment steps, where $\epsilon_{\text{eval}} = 0.05$. Furthermore, we report the mean $Q$-value at the end to study divergence. In the next section, we report the results averaged over the 6 games for 5 seeds each for a total of *30,000 GPU hours*. To combine the results of various games, we standardized the scores of each game. This involved comparing the scores obtained from playing each game with a random policy to the highest scores achieved by any of the policies analyzed. Non-standardized game results are provided in the appendix.

---

[2]Excluding Pong since every algorithm achieves near perfect score.

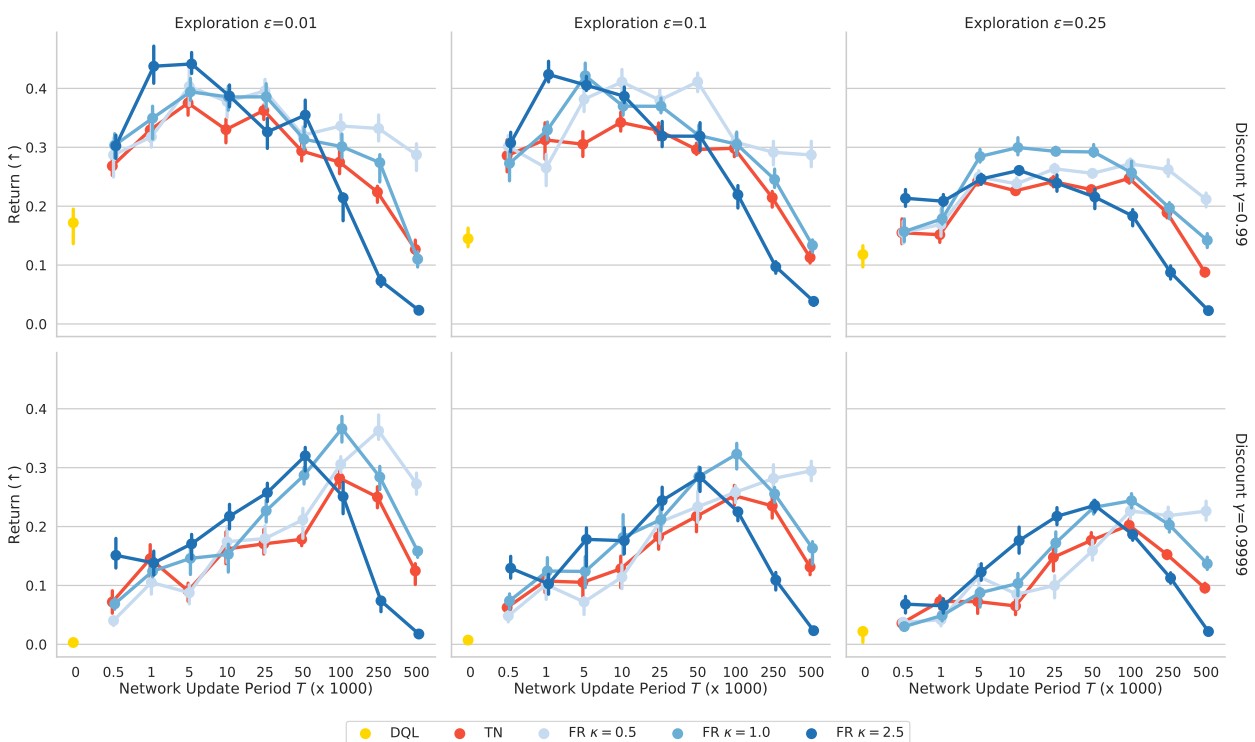

(a) **Final return for different hyper-parameters (Higher better)** We observe that increasing the exploration rate $\epsilon$ and the discount rate $\gamma$ result in performance decrease for every algorithms. However, we note that a mix of larger network update period $T$ and regularization weight $\kappa$ mitigates the performance decrease.

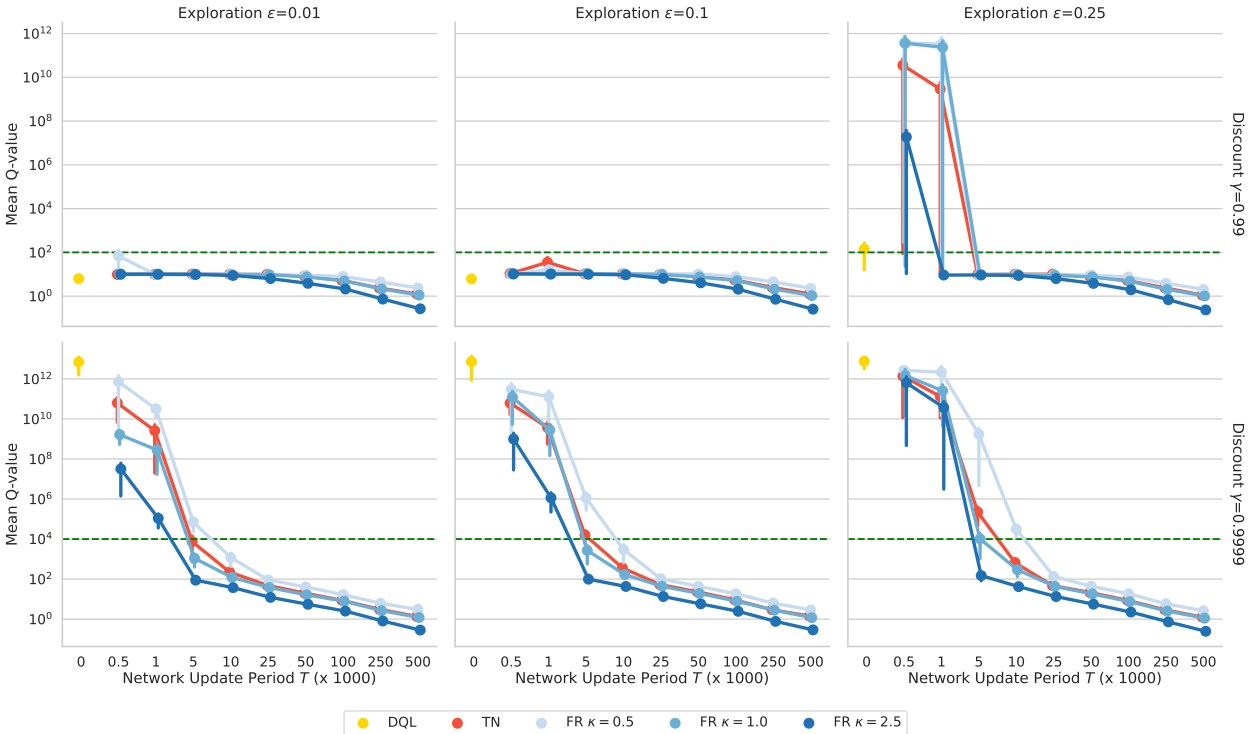

(b) **Final predicted mean $Q$ value (The green dotted line denotes divergence)** Mean Q values larger than the green line are considered diverging. We report the mean $Q$-values over the 6 environments studied. We observe that larger discount rates and more exploratory policies increase the (soft-) divergence problem. We also observe that increasing network update period $T$ or the regularization weight $\kappa$ mitigate the divergence problem.

Figure 5: Atari Ablation Study

### 4.4.2 Results

The returns and the mean $Q$-values prediction for the normalized Atari environments can be observed in Figure 5a and Figure 5b respectively.

**Increasing exploration $\epsilon$.** In Figure 5a, for discount rate $\gamma = 0.99$, we observe that for $\epsilon = 0.01$ a large regularization weight $\kappa = 2.5$ and small network update period $T$ results in the best performance. Increasing exploration $\epsilon$ to 0.1, the performance of FR is fairly similar for $\kappa \in \{0.5, 1.0\}$ and TN, but slightly worst for $\kappa = 2.5$. For $\epsilon = 0.25$, the performance of every method studied drop, and $\kappa = 1$ achieves the highest performance. Furthermore for all methods, we observe an inverted "U" shape appearing as we increase $T$ highlighting the regularization trade-off. Interestingly for smaller exploration $\epsilon$, the algorithms studied do not diverge, but for exploration $\epsilon = 0.25$ we observe that for smaller network update period $T$, vanilla DQL, TN and FR now diverge. For discount rate $\gamma = 0.9999$, we also observe that increasing the exploration $\epsilon$ past 0.1 decreases overall performance, and that the shape of the return curves exhibit a similar inverted "U" shape. As we increase $\epsilon$, for each regularization weight $\kappa$, we observe that the best performing network period update $T = 100000$ does not change. We further observe that regularization weight $\kappa$ and network period update $T$ are both effective at reducing $Q$-value divergence. For example, for $\epsilon = 0.25$ and $T = 5000$, FR $\kappa \in \{0.5, 1.\}$ diverge, while FR $\kappa = 2.5$ does not. **Overall, increasing exploration $\epsilon$ decreases performance and can result in soft-divergence of the $Q$-values.**

**Increasing the discount factor $\gamma$.** We observe that for a discount rate of $\gamma = 0.99$, both TN and FR exhibits consistent performance across a broad range of regularization values $\kappa$ and network update periods $T$. Moreover, we observe that with a discount factor of $\gamma = 0.99$ and exploration $\epsilon < 0.25$, the $Q$-values do not diverge. However, for $\epsilon = 0.25$, a combination of low regularization weight $\kappa$ and update period $T$ leads to divergence. We again note that for a fixed network update period e.g., $T = 1000$, increasing the regularization weight $\kappa$ prevent divergence. As the discount rate is increase to $\gamma = 0.9999$, we observe a consistent drop in performance. We also observe, that this higher discount rate makes the model more sensitive to regularization hyper-parameters, yielding a more pronounced inverted "U" shape that clearly illustrates the regularization trade-off. Furthermore, we observe that increasing $\gamma$ causes the $Q$-value to diverge for DQL, FR, and TN when using smaller network update periods $T$. We observe that increasing the update period $T$ and regularization $\kappa$ can prevent this divergence. **Overall, increasing the discount factor $\gamma$ can degrade performance and lead to divergence for smaller $T$. For larger $T$, we observe small degradation in performance and no divergence.**

**Overall performance.** We find that best overall performance is attained when the regularization weight $\kappa = 2.5$ and the network period update, $T$, is small. This emphasizes the significance of the additional regularization weight, $\kappa$, for achieving good results with FR. Furthermore, while their performance profile is similar, **we observe that FR with $\kappa = 1$ matches or slightly outperforms TN in all scenarios examined, indicating that FR without tuning $\kappa$ can effectively serve as a substitute for TN in the Atari suite and to match its performance or result in performance improvement.**

## 5   Related work

Multiple previous works have investigated how to improve value estimation through various constraints and regularizations. Shao et al. (2020) proposed adding an additional backward Squared Bellman Error loss to the next $Q$-value to stabilize training and remove the need for a target network on some control problems. Farahmand et al. (2009) is perhaps the closest work to our own, they regularize the reproducing kernel Hilbert space (RKHS) norm of the $Q$-value estimates, i.e., penalizing $\kappa ||Q_w||_{\mathcal{H}}^2$. However, penalizing the magnitude of the $Q$-values would prevent the algorithm from converging to $Q^*$ if $\kappa$ does not tend towards 0. Penalizing the output of a DNN draws connections with popular regularization methods in the field of policy optimization which were inspired by trust-region algorithms (Schulman et al., 2015a; Abdolmaleki et al., 2018b;a) in which the policy is KL regularized towards its past values. In the same spirit, for value-based methods, our proposed Functional Regularization regularizes the $Q$ function estimates towards their past values. Alternatively, Kim et al. (2019) showed that by using the mellow max operator as an alternative to the max operator used in bootstrapping, it was possible to stabilize training and train without a target network on some Atari games.

Other works have sought to stabilize the $Q$-value estimates by constraining parameter updates, e.g., through regularization (Farebrother et al., 2018), conjugate gradient methods (Schulman et al., 2015b), pre-conditioning the gradient updates (Knight & Lerner, 2018; Achiam et al., 2019), or using Kalman filtering (Shashua & Mannor, 2019; 2020). However, weight regularization might be ineffective in DNNs. For neural networks, the network outputs depend on the weights in a complex way and the exact value of the weights may not matter much. What ultimately matters is the network outputs (Benjamin et al., 2018; Khan et al., 2019), and it is better to directly regularize those. For instance, while Polyak's averaging (Lillicrap et al., 2015), a common weight regularization technique, has found success in control problems (Haarnoja et al., 2018; Fujimoto et al., 2018), periodically updating the parameters is usually preferred for complex DNN architectures (Mnih et al., 2013; Hausknecht & Stone, 2015; Hessel et al., 2018; Kapturowski et al., 2018; Parisotto et al., 2020).

Recently (Zhang et al., 2021) studied how Target Networks can help stabilize the TD algorithm, which is seemingly at odds with our Theorem 3.1. However, in their paper, they study a variant of DQN with TN which involves projection steps while our analysis focuses on the version used in practice.

## 6 Limitations

1. FR is not guaranteed (theoretically and empirically) to always better than TN. Indeed, for value estimation, TN can, in some cases, stabilize a divergence TD iteration while FR cannot.

2. Compared to TN, FR adds a new hyperparameter $\kappa$ and needs one additional forward propagation. However we find $\kappa$ easy to tune in practice, the default value $\kappa = 1$ being a strong baseline.

3. Our theoretical analysis is limited to the batch value estimation setting with linear function approximation. This means that we do not analyze the stochastic case where states and actions are sampled.

4. Additional research is required to study the interaction of FR with more sophisticated algorithms such as n-step return and double Q-learning.

## 7 Conclusion

In this paper, we cast a light on the implicit regularization performed by using Target Networks in RL. We have shown that despite being used as a regularizer Target Networks can cause instabilities while also being inflexible as the regularization depends on the discount factor $\gamma$. To overcome these issues, we introduced Functional Regularization which directly regularizes the value network towards a network parameterized by lagging parameters.

We performed extensive ablation studies on TN and FR to understand their behavior on a wide variety of settings. Overall decoupling the regularization parameter $\kappa$ results in a more flexible behavior and can lead to better performance. Importantly, we also observe that using the default $\kappa = 1$, FR matches or outperforms TN on our Atari experiments. This indicates that FR could be used as a drop-in replacement for TN in deep RL.

### Broader Impact Statement

This work is mostly theoretical in nature and we do not foresee direct negative societal impacts.

### Author Contributions

AP had the initial idea, implemented the experiments on Atari and 4 Rooms, and derived a first version of the link between TN and FR (Eq 5). VT derived the theory and implemented the 2-state MDP experiment. RP helped setting up the cluster to conduct the experiments. JM helped with the derivations in a previous version of the paper. GMM, CP and MEK advised on the project.

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

# A  Proofs for Linear Function Approximation case

## A.1  Supporting lemma

**Lemma A.1** (Difference of inverses). *For A and B two non-singular matrices*

$$A^{-1} - B^{-1} = A^{-1}(B - A)B^{-1}$$

*Proof.* We multiply each side by $A$ on the left and $B$ on the right and show they are equal. Left hand term:

$$A(A^{-1} - B^{-1})B = B - A$$

And for the right hand term:

$$AA^{-1}(B - A)B^{-1}B = B - A$$

By equating both terms and multiplying by $A^{-1}$ on the left and $B^{-1}$ on the right we have the equality.

$\square$

**Proposition A.2** (Convergence of TD-TN). *For $\Pi_\Phi = \Phi(\Phi^\top D\Phi)^{-1}\Phi^\top D$, the projection matrix onto the span of $\Phi$, if $\rho(\Pi_\Phi P^\pi) < \frac{1}{\gamma}$ then for $T$ large enough, Target Network Value Iteration is guaranteed to converge.*

## A.2  Proof for the domain of convergence of TD(0) with TN

The solution for the TD iteration is

$$w^* = (\Phi D(I - \gamma P^\pi)\Phi)^{-1}\Phi^\top DR \tag{8}$$

### A.2.1  Regularized iteration

Let us call $\bar{w}$ the frozen weight. We consider in this subsection only the "inner loop iteration", i.e the ones for a fixed frozen weight vector. The update for TD with target network is

$$w_{t+1} = w_t + \eta\Phi^\top D(R + \gamma P^\pi \Phi\bar{w} - \Phi w_t)$$

We call $w^*(\bar{w})$ the fixed point of that iteration rule which satisfies

$$w^*(\bar{w}) = (\Phi^\top D\Phi)^{-1}\Phi^\top D(R + \gamma P^\pi \Phi\bar{w}) \tag{9}$$

Thus our updates satisfy

$$\begin{aligned}
w_{t+1} - w^*(\bar{w}) &= w_t + \eta\Phi^\top D(R + \gamma P^\pi \Phi\bar{w} - \Phi w_t) - w^*(\bar{w}) \\
&= (I - \eta\Phi^\top D\Phi)w_t + \eta\Phi^\top D(R + \gamma P^\pi \Phi\bar{w}) - w^*(\bar{w}) \\
&= (I - \eta\Phi^\top D\Phi)w_t + \eta\Phi^\top D\Phi w^*(\bar{w}) - w^*(\bar{w}), \text{using } 9 \\
&= (I - \eta\Phi^\top D\Phi)(w_t - w^*(\bar{w}))
\end{aligned}$$

So

$$w_t - w^*(\bar{w}) = (I - \eta\Phi^\top D\Phi)^t(w_0 - w^*(\bar{w})) \tag{10}$$

As $\Phi^\top D\Phi$ is symmetric real with positive eigenvalues, $\eta \leq \frac{2}{\lambda_1}$ with $\lambda_1$ its largest eigenvalue, $w_t$ is guaranteed to converge linearly to $w^*(\bar{w})$.

### A.2.2  Distance between regularized and original optima

Now we look at the vector $w^*(\bar{w}) - w^*$

$$
\begin{aligned}
w^*(\bar{w}) - w^* &= (\Phi^\top D\Phi)^{-1}\Phi^\top D(R + \gamma P^\pi \Phi \bar{w}) - (\Phi D(I - \gamma P^\pi)\Phi)^{-1}\Phi DR \\
&= ((\Phi^\top D\Phi)^{-1} - (\Phi^\top D(I - \gamma P^\pi)\Phi)^{-1})\Phi^\top DR + (\Phi^\top D\Phi)^{-1}\gamma \Phi^\top D P^\pi \Phi \bar{w} \\
&= -\gamma (\Phi^\top D\Phi)^{-1}\Phi^\top D P^\pi \Phi(\Phi D(I - \gamma P^\pi)\Phi)^{-1})\Phi^\top DR + \gamma (\Phi^\top D\Phi)^{-1}\Phi^\top D P\Phi \bar{w}, \text{ using Lemma A.1} \\
&= -\gamma (\Phi^\top D\Phi)^{-1}\Phi^\top D P^\pi \Phi w^* + \gamma (\Phi^\top D\Phi)^{-1}\Phi^\top D P\Phi \bar{w}, \text{ By definition of } w^* \\
&= \gamma (\Phi^\top D\Phi)^{-1}\Phi^\top D P^\pi \Phi(\bar{w} - w^*)
\end{aligned}
$$

We summarize it here, this is the bias between the regularized and true optima

$$
\boxed{w^*(\bar{w}) - w^* = \gamma (\Phi^\top D\Phi)^{-1}\Phi^\top D P^\pi \Phi(\bar{w} - w^*)} \tag{11}
$$

### A.2.3  Distance between inner loop iterate and optimum

$$
\begin{aligned}
\mathcal{T}_K(\bar{w}) - w^* &= \mathcal{T}_K(\bar{w}) - w^*(\bar{w}) + w^*(\bar{w}) - w^* \\
&= (I - \eta\Phi^\top D\Phi)^K(\bar{w} - w^*(\bar{w})) + \gamma (\Phi^\top D\Phi)^{-1}\Phi^\top D P^\pi \Phi(\bar{w} - w^*) \\
&= (I - \eta\Phi^\top D\Phi)^K(\bar{w} - w^* - (w^*(\bar{w}) - w^*)) + \gamma (\Phi^\top D\Phi)^{-1}\Phi^\top D P^\pi \Phi(\bar{w} - w^*) \\
&= [(I - \eta\Phi^\top D\Phi)^K(I - \gamma (\Phi^\top D\Phi)^{-1}\Phi^\top D P^\pi \Phi) + \gamma (\Phi^\top D\Phi)^{-1}\Phi^\top D P^\pi \Phi](\bar{w} - w^*)
\end{aligned}
$$

It is possible to rewrite it in value space by multiplying by $\Phi$ on the left

$$
\begin{aligned}
\mathcal{T}_K(\bar{Q}) - Q^* &= \Phi(I - \eta\Phi^\top D\Phi)^K(I - \gamma (\Phi^\top D\Phi)^{-1}\Phi^\top D P^\pi)(\bar{Q} - Q^*) + \gamma \Phi(\Phi^\top D\Phi)^{-1}\Phi^\top D P^\pi \Phi(\bar{w} - w^*) \\
&= \Phi(I - \eta\Phi^\top D\Phi)^{K-1}((\Phi^\top D\Phi)^{-1} - \eta I)\Phi^\top D\Phi(I - \gamma (\Phi^\top D\Phi)^{-1}\Phi^\top D P^\pi \Phi)(\bar{w} - w^*) + \gamma \Pi_\Phi P^\pi(\bar{Q} - Q^*) \\
&= \Phi(I - \eta\Phi^\top D\Phi)^{K-1}((\Phi^\top D\Phi)^{-1} - \eta I)\Phi^\top D(I - \gamma \Phi(\Phi^\top D\Phi)^{-1}\Phi^\top D P^\pi \Phi)(\bar{Q} - Q^*) + \gamma \Pi_\Phi P^\pi(\bar{Q} - Q^*) \\
&= \Phi(I - \eta\Phi^\top D\Phi)^{K-1}((\Phi^\top D\Phi)^{-1} - \eta I)\Phi^\top D(I - \gamma \Pi_\Phi P^\pi)(\bar{Q} - Q^*) + \gamma \Pi_\Phi P^\pi(\bar{Q} - Q^*) \\
&= \left[ \Phi(I - \eta\Phi^\top D\Phi)^{K-1}((\Phi^\top D\Phi)^{-1} - \eta I)\Phi^\top D(I - \gamma \Pi_\Phi P^\pi) + \gamma \Pi_\Phi P^\pi \right](\bar{Q} - Q^*)
\end{aligned}
$$

The interesting bit here when analyzing how this behaves for large $K$ is that $(I - \eta\Phi^\top D\Phi)^{K-1}$ is essentially a matrix that can become arbitrarily small when $K$ is large.

$$
\mathcal{T}_\infty(Q(\bar{w})) - Q^* = \gamma \Pi_\Phi P^\pi(Q(\bar{w}) - Q^*) \tag{12}
$$

This is guaranteed to converge if $\Pi_\Phi P$ is a contraction, which is the case for instance when $D$ is the stationary distribution of $P$.

Also as $Q^* = \Phi w^*$ is the fixed point of the projected Bellman iteration, we have

$$
\mathcal{T}_\infty(Q(\bar{w})) = \Pi_\Phi\big(R + \gamma P^\pi Q(\bar{w})\big)
$$

This is the *projected value iteration* or *projected policy evaluation* algorithm (Bertsekas & Tsitsiklis, 1996).

### A.2.4   Continuity of norms and spectral radius

Now if we suppose that $\rho(\Pi_\Phi P^\pi) < \frac{1}{\gamma}$ or written differently $\rho(\gamma\Pi_\Phi P^\pi) \leq \alpha < 1$, as $\rho$ is a continuous function of its input entries, for $K$ large enough, we can guarantee that

$$\forall \epsilon > 0, \exists K \in \mathbb{N} / \rho(\Phi(I - \eta\Phi^\top D\Phi)^{K-1}((\Phi^\top D\Phi)^{-1} - \eta I)\Phi^\top D(I - \gamma\Pi_\Phi P^\pi\Phi) + \gamma\Pi_\Phi P^\pi) < \alpha + \epsilon$$

In particular for $\epsilon < 1 - \alpha$ which would ensure the convergence of the algorithm.

The lower bound on $K$ can be made more precise using the Bauer-Fike theorem $\rho(A+B) \leq \rho(A) + \|B\|\kappa(P_A)$

We can write

$$\begin{aligned}
\rho_K &\leq \gamma\rho(\Pi_\Phi P^\pi) + \kappa\|(I - \eta\Phi^\top D\Phi)^K(I - \gamma(\Phi^\top D\Phi)^{-1}\Phi^\top DP^\pi\Phi)\| \\
&\leq \gamma\rho(\Pi_\Phi P^\pi) + C\|(I - \eta\Phi^\top D\Phi)\|^K.
\end{aligned}$$

where $C = \kappa\|I - \gamma(\Phi^\top D\Phi)^{-1}\Phi^\top DP^\pi\Phi)\|$ for $\kappa$ the condition number of the eigenvector matrix of $\Pi_\Phi P$.

So we can take

$$K > \frac{\log\frac{C}{1-\gamma\rho(\Pi_\Phi P^\pi)}}{\log\frac{1}{\|(I-\eta\Phi^\top D\Phi)\|}}$$

which ensures the spectral radius of TD-TN is $< 1$.

### A.3   Proof for TD-FR, Proposition 3.2

Let us consider the case where parametrize $Q$ with a linear function approximation, $Q = \Phi w$, $\Phi \in \mathbb{R}^{|\mathcal{S}|\cdot|\mathcal{A}|\times p}$, $w \in \mathbb{R}^p$.

### A.3.1   Part 1: Fixed point of the regularized loss

Let us look at when the gradient is 0:

$$-\Phi^\top D(R + \gamma P^\pi\Phi w - \Phi w) + \kappa\Phi^\top D\Phi(w - \bar{w}) = 0$$

Thus $\Phi^\top D(I + \kappa - \gamma P^\pi)\Phi w = \Phi^\top DR + \kappa\Phi^\top D\Phi\bar{w}$. We call

$$\boxed{w_\kappa(\bar{w}) = (\Phi^\top D(I + \kappa - \gamma P^\pi)\Phi)^{-1}(\Phi^\top DR + \kappa\Phi^\top D\Phi\bar{w})}$$

the solution of the regularized problem. We denote $A_\kappa = \Phi^\top D(I + \kappa - \gamma P^\pi)\Phi$ the regularized *iteration* matrix. The TD(0) update is therefore

$$\begin{aligned}
w_{t+1} &= (I - \eta A_\kappa)w_t + \eta(\Phi^\top DR + \kappa\Phi^\top D\Phi\bar{w}) \\
&= (I - \eta A_\kappa)w_t + \eta A_\kappa w_\kappa(\bar{w})
\end{aligned}$$

Therefore

$$w_{t+1} - w_\kappa(\bar{w}) = (I - \eta A_\kappa)(w_t - w_\kappa(\bar{w})) \tag{13}$$

### A.3.2   Part 2: Difference between regularized optima and true optima

Let us look at the quantity $w_\kappa(\bar{w}) - w^*$ where $w^* = A_0^{-1}\Phi^\top DR$, the solution of the unregularized problem. First, we will use the fact that $A^{-1} - B^{-1} = A^{-1}(B - A)B^{-1}$ to get

$$
\begin{aligned}
w_\kappa(\bar{w}) - w^* &= A_\kappa^{-1}(\Phi^\top DR + \kappa\Phi^\top D\Phi\bar{w}) - A_0^{-1}\Phi^\top DR \\
&= -\kappa A_\kappa^{-1}\Phi^\top D\Phi A_0^{-1}\Phi^\top DR + \kappa A_\kappa^{-1}\Phi^\top D\Phi\bar{w} \\
&= \kappa A_\kappa^{-1}\Phi^\top D\Phi(\bar{w} - w^*)
\end{aligned}
$$

Thus we have

$$
w_\kappa(\bar{w}) - w^* = \kappa A_\kappa^{-1}\Phi^\top D\Phi(\bar{w} - w^*) \tag{14}
$$

### A.3.3   All together

$$
\begin{aligned}
\mathcal{T}_\kappa^T(w_t) - w^* &= \mathcal{T}_\kappa^T(w_t) - w_\kappa(w_t) + w_\kappa(w_t) - w^* \\
&= (I - \eta A_\kappa)^T(w_t - w_\kappa(w_t)) + \kappa A_\kappa^{-1}\Phi^\top D\Phi(w_t - w^*), \text{ using 13 and 14} \\
&= (I - \eta A_\kappa)^T(w_t - w^* - (w_\kappa(w_t) - w^*)) + \kappa A_\kappa^{-1}\Phi^\top D\Phi(w_t - w^*) \\
&= (I - \eta A_\kappa)^T(w_t - w^* - \kappa A_\kappa^{-1}\Phi^\top D\Phi(w_t - w^*)) + \kappa A_\kappa^{-1}\Phi^\top D\Phi(w_t - w^*), \text{ Using 14 again} \\
&= \left[(I - \eta A_\kappa)^T(I - \kappa A_\kappa^{-1}\Phi^\top D\Phi) + \kappa A_\kappa^{-1}\Phi^\top D\Phi\right](w_t - w^*)
\end{aligned}
$$

### A.3.4   Studying $I - \eta A_\kappa$

For this, let us study the eigenvalues of $A_\kappa$, and make sure they have a positive real part. Then by choosing $\eta \leq 2\,\mathfrak{Re}(\frac{1}{\lambda_{\max}})$ the matrix will be stable.

For this, we will need, an assumption, i.e that TD(0) converges for that combination of $\Phi, D, P$ and $\gamma$.

**Assumption A.3.** $Sp(\Phi^\top D\Phi - \gamma\Phi^\top DP^\pi\Phi) \subset \mathbb{C}^+$

Let us call $\epsilon_\kappa = \min_{\lambda \in Sp(A_\kappa)} \mathfrak{Re}(\lambda)$.

As $\lim_{\kappa \to 0^+} A_\kappa = \Phi^\top D\Phi - \gamma\Phi^\top DP^\pi\Phi$, we have $\epsilon_0 > 0$ and by continuity of the spectrum, $\exists\delta$ so that $\forall\kappa < \delta$, $\epsilon_\kappa \geq \frac{\epsilon_0}{2} > 0$. Thus we just showed that for $\kappa$ small enough the matrix $A_\kappa$ was also stable.

### A.3.5   Studying $\kappa A_\kappa^{-1}\Phi^\top D\Phi$

Here we just need to show that for $\kappa$ small enough this matrix will be stable, i.e have a spectral radius bounded by 1. It is sufficient to notice that $A_\kappa^{-1}\Phi^\top D\Phi$ has a finite limit when $\kappa \to 0$, which is equal to $A_0^{-1}\Phi^\top D\Phi$ which is indeed bounded as $A_0 = \Phi^\top D\Phi - \gamma\Phi^\top DP^\pi\Phi$ has non-zero eigenvalues as they are $\subset \mathbb{C}^+$ as per the Assumption above.

Thus, for $\lim_{\kappa \to 0} \kappa A_\kappa^{-1}\Phi^\top D\Phi = 0$ which has a spectral radius of 0. Then again, by continuity of the spectrum thus spectral radius, for $\kappa$ small enough, $\kappa A_\kappa^{-1}\Phi^\top D\Phi$ has a spectral radius $< 1$.

**Corollary A.4** (Stability of TD-FR and TD-TN). *When $p = 1$, for $T$ large enough, the convergence domain of TD-TN is included in the one of FR-TN.*

### A.4   Proof for Corollary A.4

We first need to show that $Sp(\Phi^\top D\Phi - \gamma\Phi^\top DP^\pi\Phi) \subset \mathbb{C}^+$ implies that $\rho\big((\Phi^\top D\Phi)^{-1}\Phi^\top DP^\pi\Phi\big) < \frac{1}{\gamma}$.

Let us consider $\mathbf{v}$ a eigenvector of $(\Phi^\top D\Phi)^{-1}\Phi^\top DP^\pi\Phi$ associated with the eigenvalue $\lambda$. Thus

$$
\Phi^\top DP^\pi\Phi\mathbf{v} = \lambda\Phi^\top D\Phi\mathbf{v}
$$

From this we multiply by $\gamma$ and substract and add $\Phi^\top D\Phi\mathbf{v}$:

$$\left(\Phi^\top D\Phi - \gamma\Phi^\top DP^\pi\Phi\right)\mathbf{v} = (1 - \lambda\gamma)\Phi^\top D\Phi\mathbf{v}$$

Multiplying by $\mathbf{v}^\top$:

$$\mathbf{v}^\top\left(\Phi^\top D\Phi - \gamma\Phi^\top DP^\pi\Phi\right)\mathbf{v} = (1 - \lambda\gamma)\mathbf{v}^\top\Phi^\top D\Phi\mathbf{v}$$

As $\Phi^\top D\Phi$ is positive definite (non-singular by assumption), $\mathbf{v}^\top\Phi^\top D\Phi\mathbf{v} > 0$. However we can't conclude anything in the general case with non-symmetric matrices. When $p = 1$, these matrix products become real scalars as all the coefficients are real. So for $x^2 = \Phi^\top D\Phi$

$$\left(\Phi^\top D\Phi - \gamma\Phi^\top DP^\pi\Phi\right) = (1 - \lambda\gamma)x^2$$

so $\left(\Phi^\top D\Phi - \gamma\Phi^\top DP^\pi\Phi\right) > 0$ implies that $(1 - \lambda\gamma) > 0$, thus $\lambda < \frac{1}{\gamma}$ the maximum and only eigenvalue, thus the spectral radius of $(\Phi^\top D\Phi)^{-1}\Phi^\top DP^\pi\Phi$ is smaller than $\frac{1}{\gamma}$.

We denote by $\mathcal{D}_{\text{TD}} = \{\Phi, D | Sp(\Phi^\top D\Phi - \gamma\Phi^\top DP^\pi\Phi) \subset \mathbb{C}^+\}$ and $\mathcal{D}_{\text{TN}} = \{\Phi, D | \rho\left((\Phi^\top D\Phi)^{-1}\Phi^\top DP^\pi\Phi\right) < \frac{1}{\gamma}\}$ the convergence domains of TD and TN ($T = +\infty$). We just proved for $p = 1$ that $\mathcal{D}_{\text{TN}} \subset \mathcal{D}_{\text{TD}}$. We know from Proposition A.2 and 3.2 that for $\kappa$ small enough, FR(T) and TN(T) (FR and TN using a period update of $T$) and $\mathcal{D}_{\text{FR(T)}}$, $\mathcal{D}_{\text{TN(T)}}$ their convergence domains that $\lim_{T\to+\infty}\mathcal{D}_{\text{FR(T)}} = \mathcal{D}_{\text{TD}}$ and $\lim_{T\to+\infty}\mathcal{D}_{\text{TN(T)}} = \mathcal{D}_{\text{TN}}$. Thus by taking the limit we have the inclusion we needed.

## B   Time comparison

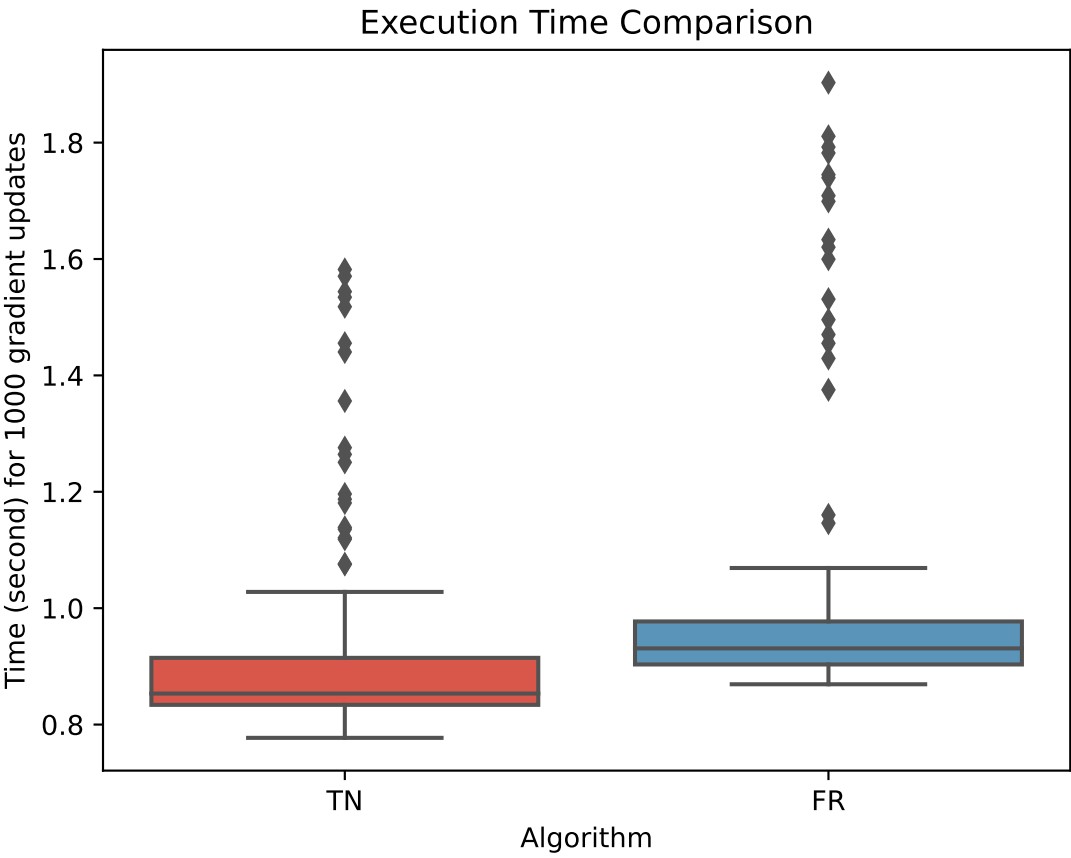

Figure 6: **Time comparison of the gradient updates between FR and TN.** We note that in a training scenario a large portion of the time is spent on inference and simulation which make the difference in gradient update time negligible.

## C   Additional Results and Hyper-parameters

### C.1   Four Rooms

Table 1: Four Rooms Hyper-parameters

| Hyperparameter | Value |
| --- | --- |
| learning rate | 1e-4 |
| optimizer | adam (Kingma & Ba, 2014) |
| discount factor $\gamma$ | 0.99 |
| DNN layers | [128, 128, 4] |
| dimension | $11 \times 11$ |

## C.2 2-state MDP

We see on Appendix C.2 that increasing the discount factor leads to larger divergence regions. However, the circle of radius $\pi(s_0)$ which corresponds to the on-policy algorithm stays in the convergent domain as predicted by theory.

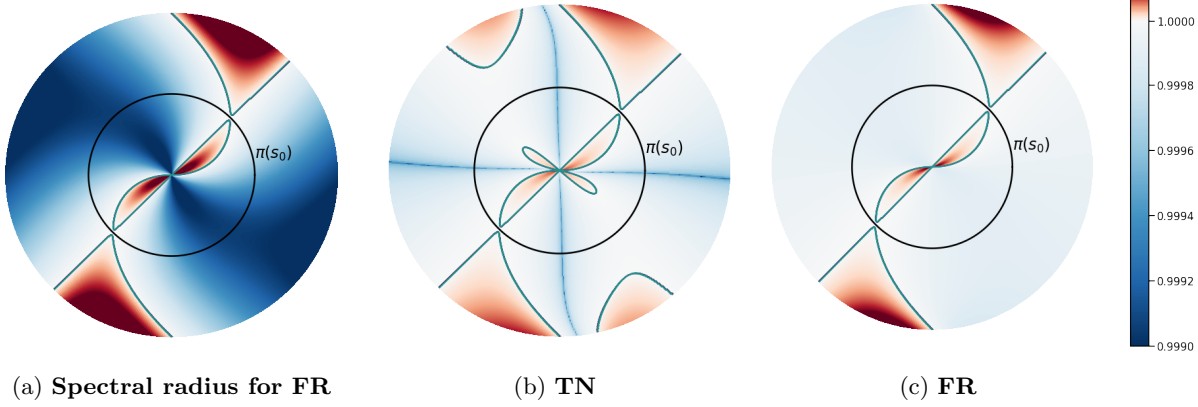

(a) **Spectral radius for FR**       (b) **TN**       (c) **FR**

Figure 7: **2-state MDP.** $\gamma = 0.9995$

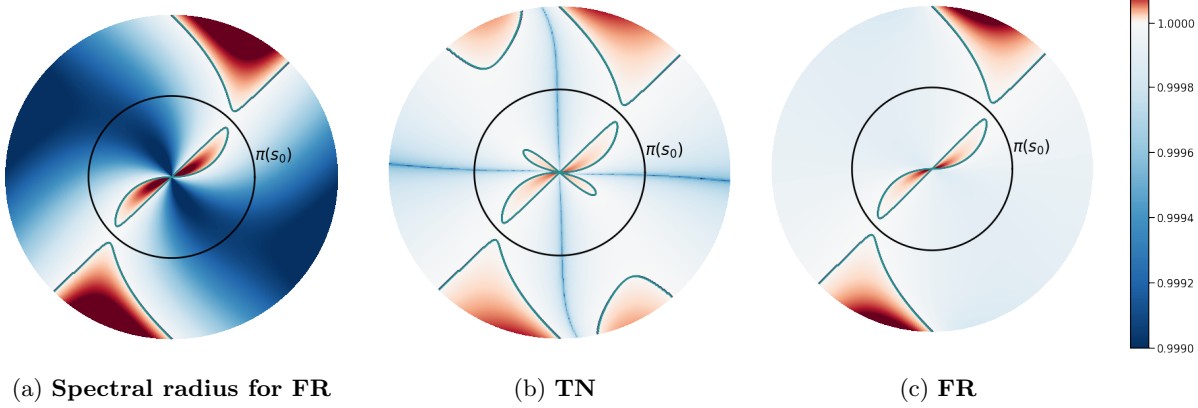

(a) **Spectral radius for FR**       (b) **TN**       (c) **FR**

Figure 8: **2-state MDP.** $\gamma = 0.995$

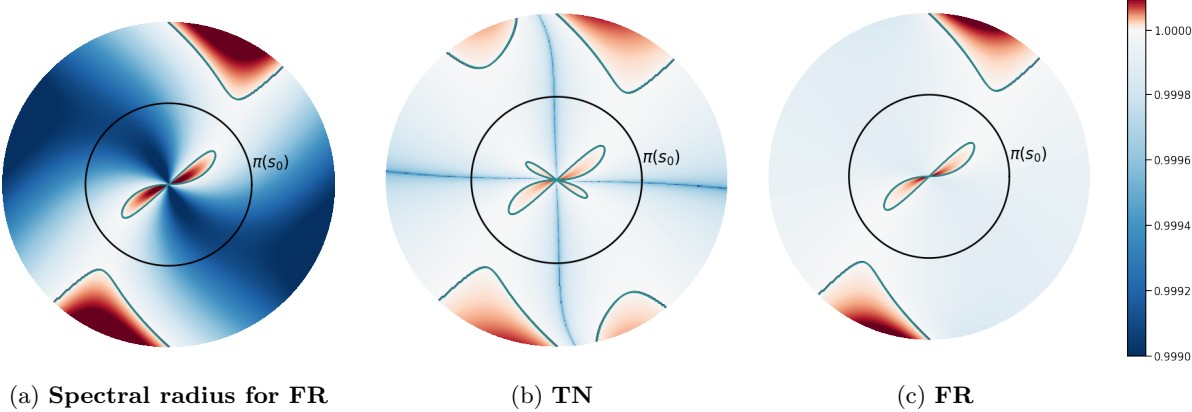

(a) **Spectral radius for FR**       (b) **TN**       (c) **FR**

Figure 9: **2-state MDP.** $\gamma = 0.95$

## C.3 Atari Suite

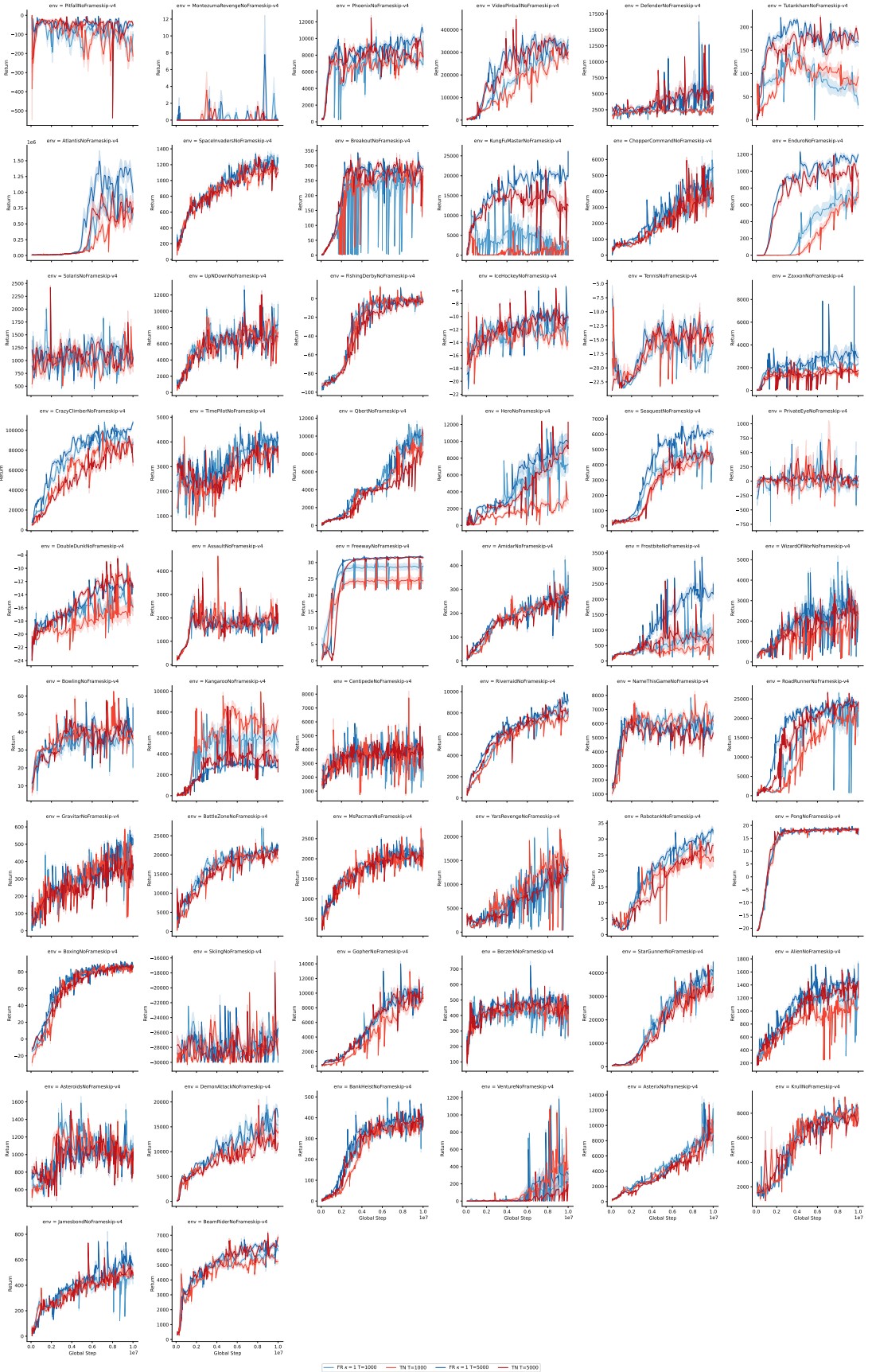

Figure 10: **Atari suite learning curves.**

## C.4 Atari Sensitivity Analysis

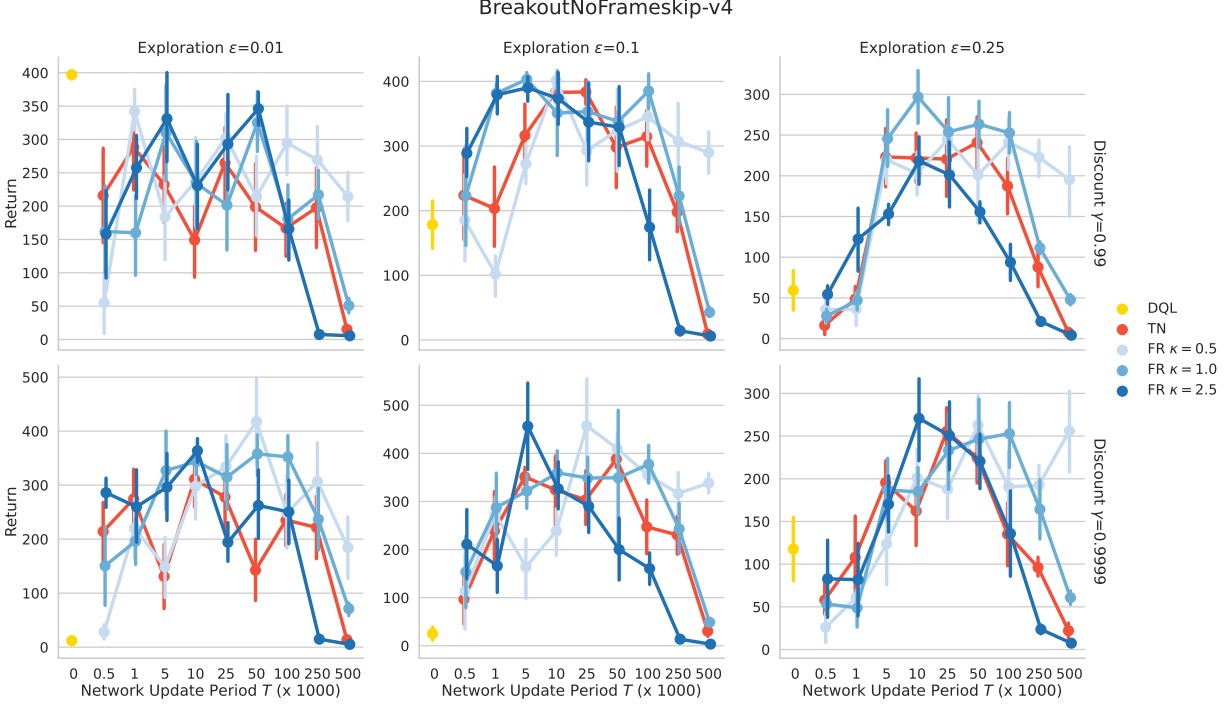

Figure 11: Breakout results

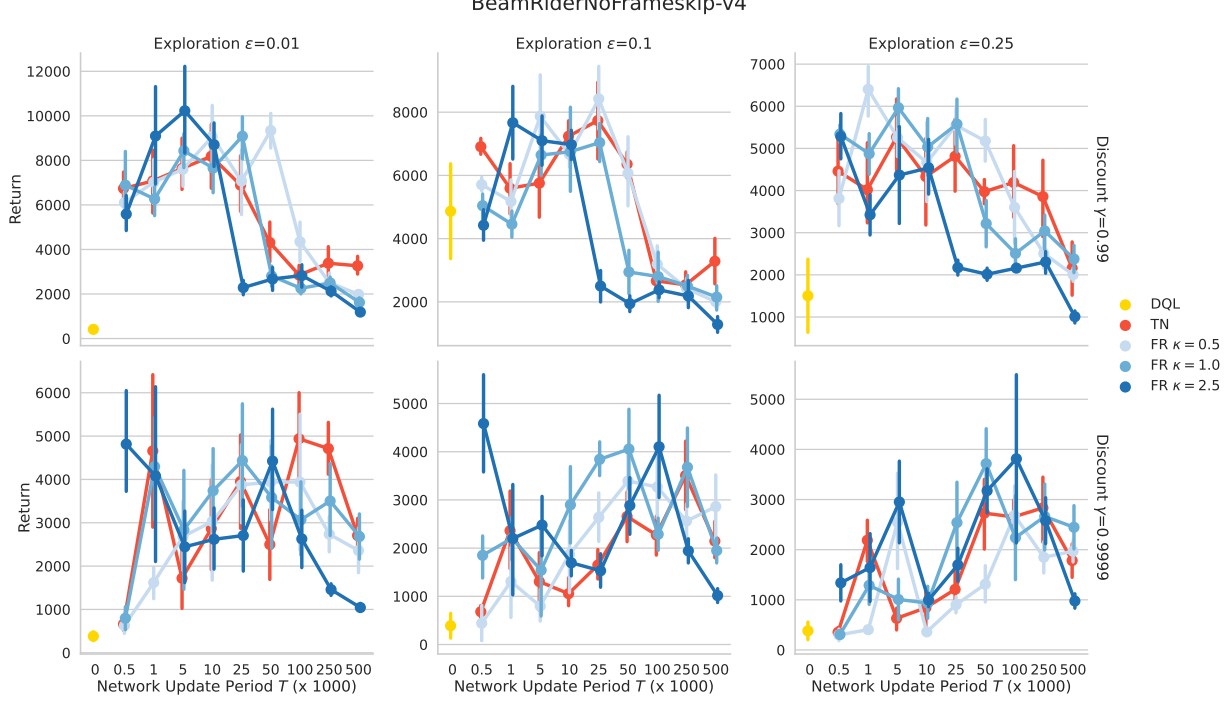

Figure 12: Beam Rider results

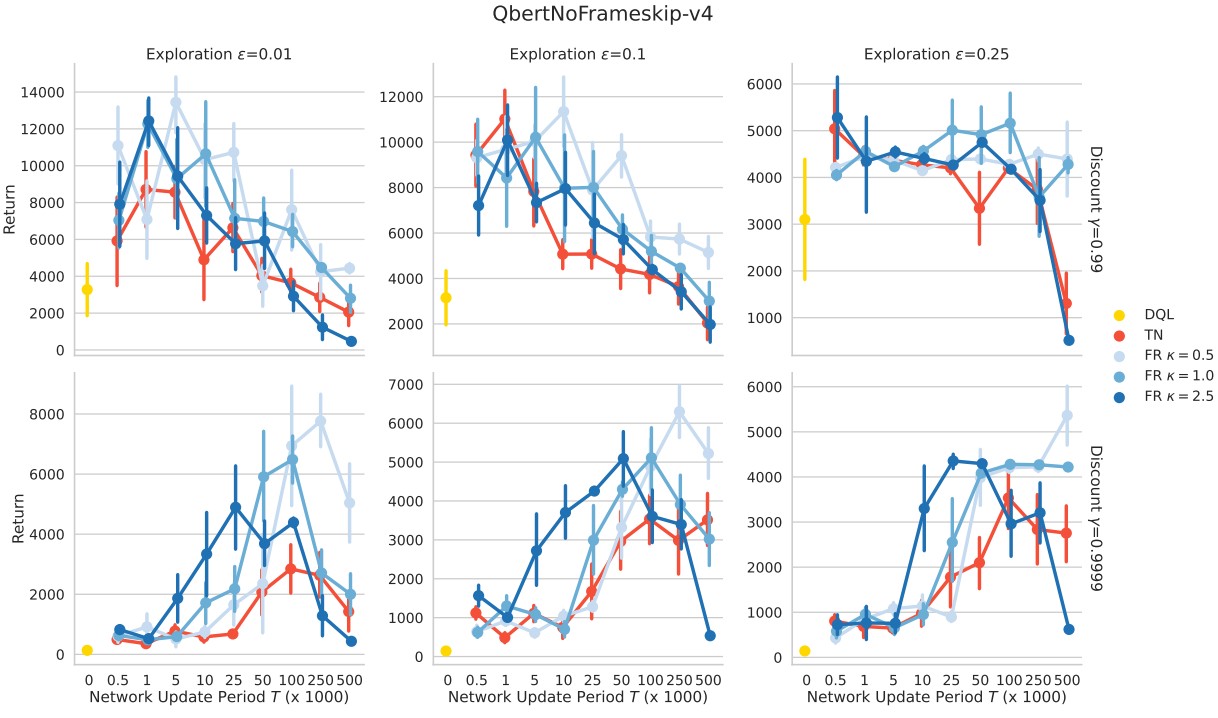

Figure 13: Qbert results

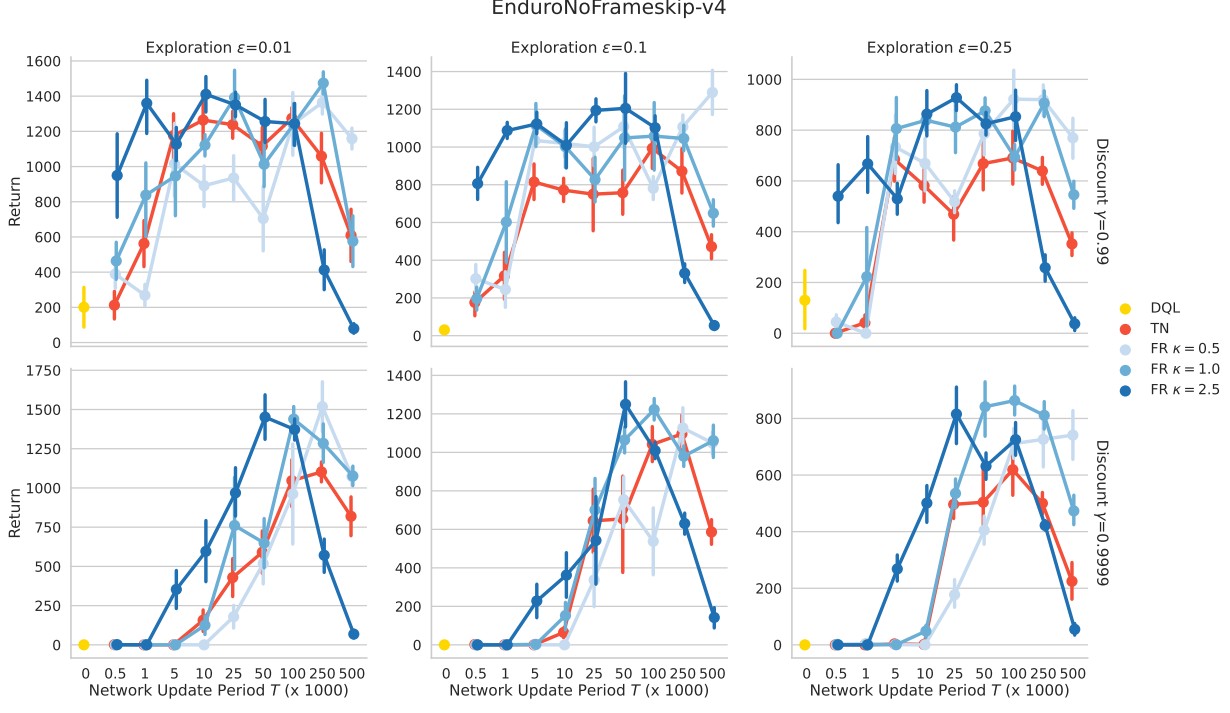

Figure 14: Enduro results

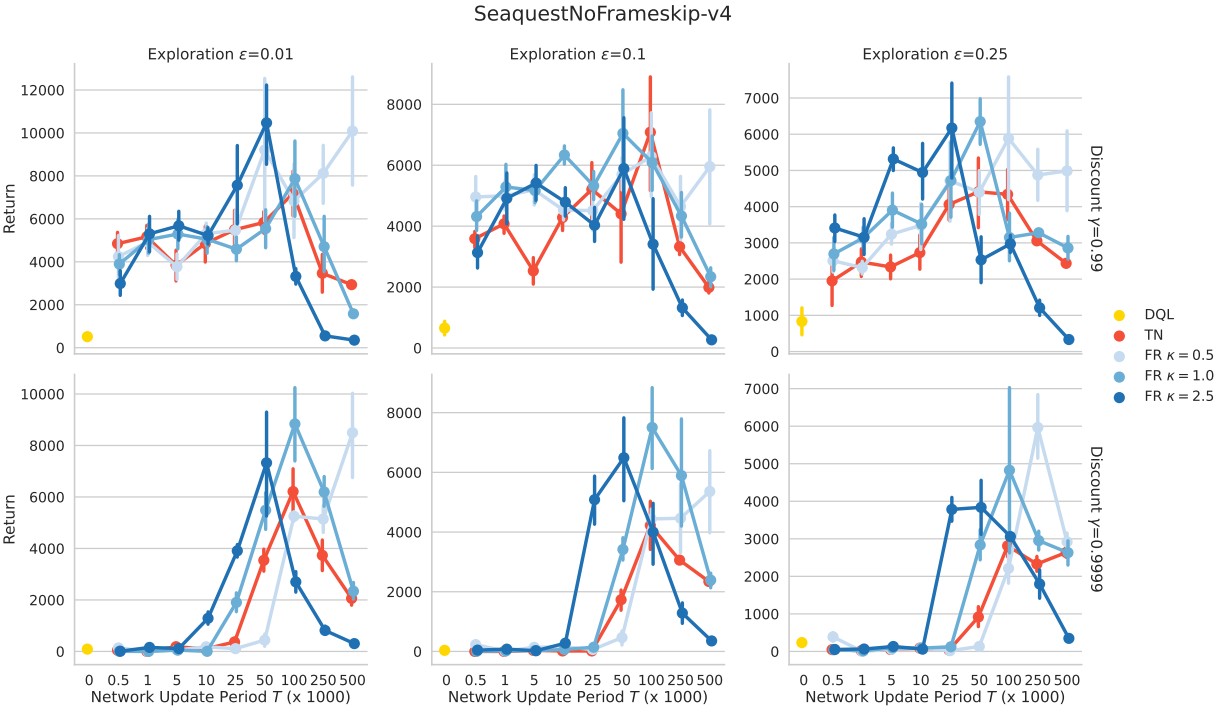

Figure 15: Seaquest results

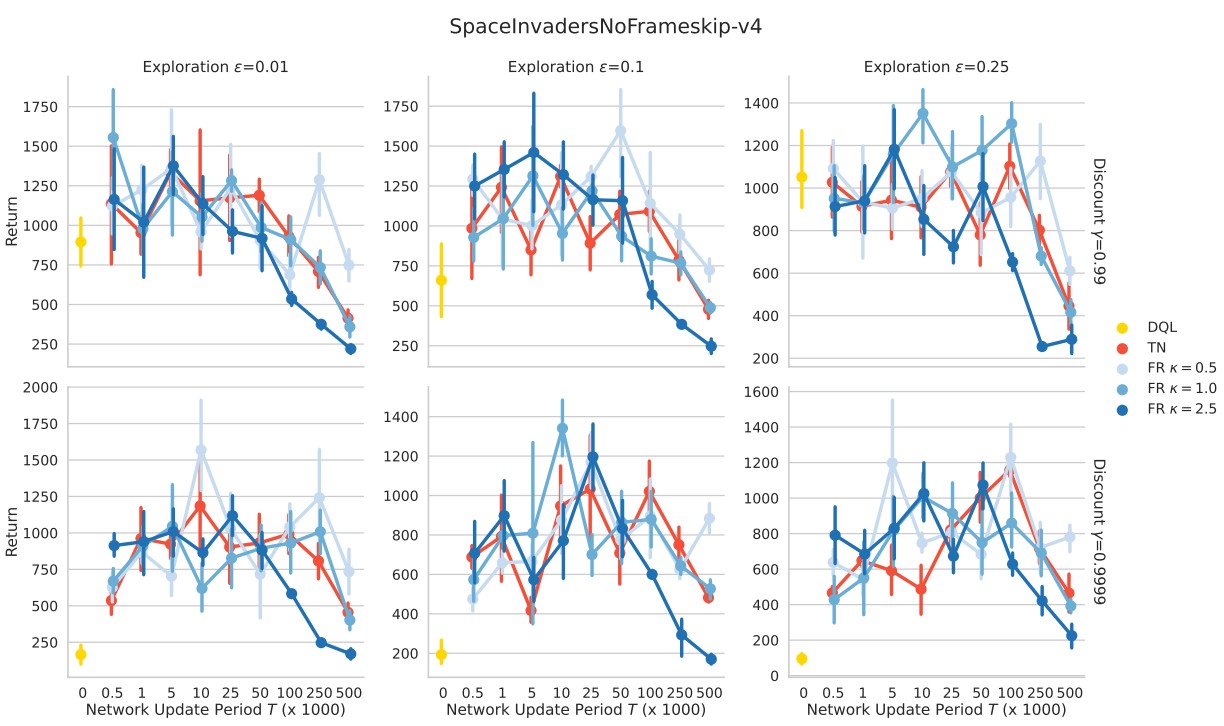

Figure 16: SpaceInvaders results

