# OpenReview forum: "Bridging the Gap Between Target Networks and Functional Regularization"
_TMLR — Accepted by TMLR_

### Review · Reviewer_nz5F · 2023-07-09

**Summary Of Contributions:**

The authors show that target network (TN) in DRL serves as an implicit regularizer that can have benefits but also has its own issues. The authors then propose an explicit functional regularization (FR) with theoretical support for its convergence. The authors compared TN and FR on classic toy example environments, and provided some interesting analysis. Further empirical results are provided for tasks in Atari, showing that FR can do better in terms of performance and prevent divergence.

**Audience:**

Yes

**Claims And Evidence:**

Yes

**Requested Changes:**

Suggestions:

- please more explicitly report computation overhead and compare to TN.
- more experiments over more tasks (maybe just pick the best FR setting and apply to other environments) can be interesting, though what are in the paper now already make some interesting contribution.

Typos: please go through the paper and fix typos, including:
- end of page 2 "We denote the the off-policy sampling"
- page 6 "disk or radius 1"
- page 7 "Figure 2 where can see"


**Strengths And Weaknesses:**

Strengths:

**Novelty**
- novel theoretical results
- new analysis on classic environments
- new results for Atari showing the FR can perform as well as TN, and in some cases stronger.

**Quality**
- Overall the presentation is good, the analysis on the toy examples are especially interesting.
- figures and results are presented and explained well, highlight in alg 1 is great.
- empirical study spans a wide range of FR hyperparameter and environment settings

**Clarity**
- the paper is written in a clear way, there are many interesting results, but is overall easy to follow

**Significance**
- the theoretical results plus the classic environment analysis can make some significant contribution
- empirical results on Atari are not very strong, but also interesting

Weaknesses:
- Additional technical details: Regarding computation overhead, how is FR compared to using TN? I guess it is not too big but would like to know the exact number. (sth like a computation time table can be good)
- some typos but can be easily fixed
- the results on Atari are over a limited number of environments
- on Atari, seems to me there are some performance gain, however, this is when looking at a number of different FR settings, and some of them do better, while others might do worse in cases. (I guess FR brings in more flexibility with the extra hyperparameter)
- Atari experiments are compared to DQN which is an older baseline and not with SOTA methods. (though the main goal of the paper is not to improve over SOTA)

An overall solid paper with some nice contributions.

---

> ### Author Response · Authors · 2023-07-27
> **Answer Reviewer nz5F**
>
> Thank you for your review, we appreciate your suggestions and used them to improve our paper.
>
> ----
>
> > please more explicitly report computation overhead and compare to TN.
>
> We added a figure to compare the computation overhead in the appendix.
>
> ----
>
> > more experiments over more tasks (maybe just pick the best FR setting and apply to other environments) can be interesting, though what is in the paper now already makes some interesting contributions.
>
> Very good point. We added experiments on the full Atari suite using kappa=1.

---

### Review · Reviewer_pCv4 · 2023-07-10

**Summary Of Contributions:**

The authors analyse TD(0) with a target network and show that it can in some cases cause divergence.

They propose an alternative solution that solves this issue using function regularization, and perform several experiments to show its performance.

**Audience:**

Yes

**Claims And Evidence:**

Yes

**Requested Changes:**

All of the following would strengthen the work, and are not critical for securing my recommendation.

1. If possible, please add to the supplementary the convergence plots of each Atari game (rewards of roll-out episodes) for the best set of parameters. It usually gives better perspective on which algorithm is better.

2. If you manage to find a theoretical connection between kappa, gamma and T it would largely contribute to the paper.

3. I would mention some of the related works along the paper in relevant locations.



**Strengths And Weaknesses:**

+ The paper is very well written.

+ The plots are clear and visually pleasing.

+ The paper has potentially high impact and is of interest to the community. I agree with the authors that FR is both simple, and also makes more sense than the currently used TN. I see no reason it shouldn't replace TNs in the common packages (say with kappa=1).

- The limitations mentioned by the authors themselves. I think the biggest issue for me is the extra parameter kappa, as it feels like some theoretical connection between kappa, gamma and T should exist.

- Several works were mentioned in the related work section, but I feel like the comparison to the closer 1 or 2 works should've been included somehow in the paper (in the experiments, or the theory). Otherwise the work feels a bit secluded.

---

> ### Author Response · Authors · 2023-07-27
> **Answer to Rev pCv4**
>
> Thank you for your review, we appreciate your suggestions and used them to improve our paper.
>
> ----
>
> > If possible, please add to the supplementary the convergence plots of each Atari game (rewards of roll-out episodes) for the best set of parameters. It usually gives a better perspective on which algorithm is better.
>
> We added the learning curves in the appendix.
>
> ----
>
> > I would mention some of the related works along the paper in relevant locations.
>
> We added a paragraph about the connection with gradient TD methods. Let us know if there is specific related work you believe would be appropriate to cite.
>
> ----
>
> > If you manage to find a theoretical connection between kappa, gamma and T it would largely contribute to the paper.
>
> Yes, this is a very sound and natural suggestion. We have looked into finding such a relationship, however we do not believe there exists one in the general case. In some sense, a difference between TN and FR is that the former adds a $\gamma P$ change to an iteration matrix, while the latter adds $\kappa I$. When the MDP is trivial and $P=I$ we have a direct equivalence between FR and TN, but generally, even factoring another degree of freedom in the choice of $T$, a relationship between $\gamma, \kappa$ and $T$ would also depend on the eigendecomposition of $P$. Thus we do not believe we can prove a useful or particularly insightful relationship.

---

> > ### Comment · Reviewer_pCv4 · 2023-07-30
> > **Thank you for updating the paper**
> >
> > I have read all reviews and rebuttals and my concerns have been addressed.
> >
> > Specifically, I am very pleased the authors included the convergence plots for all Atari games.
> > It is now clearly visible in multiple games their method outperforms the vanilla method (Enduro, Atlantis, Kung-Fu master, Hero, SeaQuest, Freeway, Robotank, DemonAttack). More strongly, its apparent that in no game the results are worse, suggesting FR is strictly better.

---

### Review · Reviewer_uvyZ · 2023-07-11

**Summary Of Contributions:**

This paper investigates the implicit regularization role of Target Networks in deep Reinforcement Learning and points out that Target Networks may lead to instabilities and inflexibility. To address these issues, the authors introduce a Functional Regularization method that directly regularizes the value network towards a network parameterized by lagging parameters. By conducting extensive ablation studies on Target Networks and Functional Regularization across various settings, the paper demonstrates that decoupling the regularization parameter κ results in more flexible behavior and improved performance. The paper conducts several experiments to verify the effectiveness of the proposed Functional Regularization method and compares it with the traditional Target Network method.

**Audience:**

Yes

**Broader Impact Concerns:**

None at this moment.

**Claims And Evidence:**

No

**Requested Changes:**

Please see weaknesses.

**Strengths And Weaknesses:**

[+] This paper analyzes the Target Network's implicit regularization from a theoretical point of view. The analysis is enlightening and reveals the flaws of the Target Network method based on theoretical analysis. The authors utilize the results of theoretical analysis to propose optimization goals and design a new algorithm.
[+] The introduced Functional Regularization method exhibits flexibility, allowing for improved performance by decoupling the regularization parameter κ.
[+] The presentation is clear and easy to follow. The experiment results are well discussed and the figures, such as Figure 1, are well organized and informative.
[-] The logic of the paper is: TN method can be interpreted as a kind of implicit regularization, but this implicit regularization has its own problems, so the authors propose a more direct regularization method. My doubts lie in: 1. The authors have not been able to theoretically explain why such regularization would be helpful for TD learning, so the logical argument that their method benefits TD learning through better regularization is questionable. 2. Perhaps the authors could observe the regularization effect in experiments, especially in general benchmark experiments through some ablation studies, to match the article's logic, rather than just directly observing the final experimental results. Because even if the experimental results improve, the logic in the previous text (FR improves TD learning through better regularization) cannot be verified.
[-] While the theoretical part of the paper is limited to a very simple setting, which is linear function approximation with finite state and action spaces, and no exploration effect is considered, the effectiveness of the proposed method should be evaluated carefully using experimental methods. However, only two simple environments show that it outperforms the TN method, while in Atari games, it is not significantly better than the baseline method. Also, the Atari experiments do not seem to have converged for all algorithms.
[?] In Four Rooms and Atari games experiments, all returns are very wild since they all drop at the end, even worse than the initial return. Can you explain why?

---

> ### Author Response · Authors · 2023-07-27
> **Answer to uvyZ**
>
> Thank you for your review, we appreciate your suggestions and used them to improve our paper.
>
> ----
>
> > Also, the Atari experiments do not seem to have converged for all algorithms.
>
> > In Four Rooms and Atari games experiments, all returns are very wild since they all drop at the end, even worse than the initial return. Can you explain why?
>
> >  Perhaps the authors could observe the regularization effect in experiments, especially in general benchmark experiments through some ablation studies
>
> The figures that were in the main paper (now Fig 3 and Fig 5) are robustness/ablation curves. Specifically, the x-axis represent different network update period and the shade of blue the regularization strength. We can observe TN in red and TD in yellow.
>
> To monitor at the convergence and learning, one can look at the learning curves are in the appendix (Fig 10).
>
> ----
>
> > The authors have not been able to theoretically explain why such regularization would be helpful for TD learning, so the logical argument that their method benefits TD learning through better regularization is questionable.
>
> It is likely that in order to fully understand the impact of FR on TD learning our theoretical analysis must be extended to include DNNs which is an active area of research. And the same thing could be said about TN.
>
> We believe that the regularization benefits of our method on TD learning are quite clear experimentally. For example, we can clearly see the impact of regularization on performance, e.g. Fig 3 a) we can see that too much or too little regularization degrades performance. We can also see that TD (in yellow) achieves much lower performance than FR or TN with tune regularization.
>
> Overall, we believe that our paper sheds is a step towards understanding more how regularization can prevent TD from diverging.
>
> ----
>
> > However, only two simple environments show that it outperforms the TN method, while in Atari games, it is not significantly better than the baseline method.
>
> We added experiments on the full Atari suite and confidence intervals on the results (Fig 4).

---

### Review · Reviewer_GKoB · 2023-07-12

**Summary Of Contributions:**

This paper proposes a perspective of viewing target networks as a form of regularization for value function learning with bootstrapping. Through this perspective, the paper shows that target networks can introduce new regions of instability for TD learning methods---counter to conventional wisdom that target networks assist with stability. The paper highlights this instability is due to nonconvexity of the regularization problem. A simple modification to target networks allows for a (sequence of) convex loss(es) which the paper shows adds no novel instability regions---though also does not improve stability. These findings are shown both theoretically and empirically in the off-policy linear function approximation case in expectation. The paper concludes with an empirical study with nonlinear function approximation on a small set of Atari games and a wide range of hyperparameter settings for the proposed method.

**Audience:**

Yes

**Broader Impact Concerns:**

No broader impact concerns

**Claims And Evidence:**

No

**Requested Changes:**

1) Discuss the relationship between functional regularization and gradient TD --- at the very least discussing that both share the same goals, but preferably also the significant overlap between TDC and FR.
2) Concretely discuss how FR relates to and improves stability. The results (both theoretical and empirical) in the linear FA section don't seem to support this claim.
3) Consider a strategy to reduce hyperparameter sensitivity of $\kappa$---the atari results required a sweep over a wide range of this hyperparameter and the hyperparameter seems strongly problem-dependent. Is it possible that defining a new hyperparameter $\eta$ such that $\kappa = \eta \gamma$ is much less sensitive?
4) Align the Atari claims with the evidence---either tone down the claims to say that FR is at least as good as TN or ramp up the evidence to suggest FR is truly superior. As an aside, the paper does not _need_ superiority claims to be impactful so toning down the claims seems the easiest path to me!

**Strengths And Weaknesses:**

Strengths:

1) The theoretical results appear sound and depend on sensible assumptions.
2) The paper is exceptionally clear. The motivation is sound, interesting to a broad audience, and timely.
3) The empirical results in Fig 1 are the best that I've seen motivating the increased potential instability from using target networks, while also clearly highlighting the loss of sample efficiency. Also these are quite beautiful plots!
4) The sensitivity curves in Fig 3 clearly highlight that the proposed algorithm dominates target networks for a very wide range of configurations of the proposed regularizer. Results are clearly statistically significant with 40 samples per condition.
5) The proposed modification is simple to implement and easy to understand. The only added cost is an additional forward pass through the NN during learning, which is negligible compared to the cost of backprop.


Weaknesses:

1) There is an unfortunate limitation that theoretical results are exclusive to the linear setting, while target networks are almost exclusively used in nonlinear settings. This deviation nullifies one of the major motivations of the proposed algorithm.
2) The paper would benefit from a richer discussion of placement in the literature. Specifically, there is no mention of gradient TD methods, and these share nontrivial similarities to the proposed approach.
3) There seems to be a subtle contradiction in motivation and the proposed algorithm. Much of the motivation is around stable learning and the proposed modification to target networks is to avoid instability due to the matrix $\Phi DP \Phi$. However, the proposed algorithm is only provably stable under the same conditions as TD, where we've already placed strong conditions on the highly related matrix $\Phi D (I - P) \Phi$.
4) The additional hyperparameters are implied to be a strength---i.e. increasing configurability of the method. I tend to view additional problem-dependent configuration parameters as a weakness, especially when it appears there is a natural dependency on observable problem information (i.e. $\gamma$). Given that the paper suggests $\kappa = 1$ is generally a good choice, which is approximately $\kappa = \gamma$, it seems we could eliminate a configuration parameter by making _fewer_ deviations from target networks.
5) The paper claims superiority on Atari, but at best the experiments provide evidence of equivalency. Because the proposed method has an additional hyperparameter, and because no single setting of that hyperparameter performs consistently well (looking in appendix B.3) it seems any marginal superiority per-game is due to additional degrees of freedom and loose confidence intervals. These claims should be toned down, or the evidence ramped up (e.g. by tightening confidence intervals which is likely outside of budget, or reducing configuration degrees of freedom).


Minor nitpicks:
- Eqn 3 really isn't the gradient of any loss function. Writing it as such on the left side of the equals sign isn't quite correct, and really isn't necessary to what is being communicated.
- At the top of page 3, it is stated that linear functions provide a poor approximation to $Q$ in complex environments. This isn't really true---consider that a neural network can be decomposed into a feature-generating function up to the penultimate layer and a linear function of those features in the last layer. If someone handed me that feature-generating function ahead of time, I could learn a linear approximation of equivalent quality as the neural network. Consider also Machado et al. 2018 that showed Atari can be easily solved with linear function approximation of the right features. Really DNNs are useful when high-quality features are not readily available and must be learned, and has little to do with environment complexity.
- I'm not sure Fig 2 provides any novel insight, and the related paragraph in the middle of page 7 might be overclaiming its novelty / contribution. It's been quite well-established that $\gamma$ plays a strong role in the divergence of TD (e.g. Kolter 2011, Scherrer 2010, White 2017)

-----

Further details:

Weakness 1 - The deviation between linear and nonlinear settings does significantly limit the insights we can draw. The dependency on linear function approximation is also nontrivial and it is unclear whether any of the insights hold beyond the linear case. One of the major contributions of the proposed algorithm is to modify the matrix $\Phi DP \Phi$ to become $\Phi D \Phi$ in order to have a convex loss. In the linear case, this is totally sensible. In the nonlinear case, target networks change both the mapping from features to value function (i.e. $w$) _and_ the feature matrix (i.e. $\Phi$).

If we look at the learning system at a given timestep, freezing the neural network and target network weights, then we can produce a "feature matrix" by feeding each state into the respective neural networks and storing the outputs of the penultimate layer. The neural network would produce some matrix $\Phi$ which is $|S| \times d$ and the target network would produce some $\Phi_{tn}$ of same dimensions. Now our regularizer term under target networks has matrix $\Phi DP \Phi_{tn}$. Removing the stochastic matrix $P$ is not enough to ensure convexity---i.e. $\Phi D \Phi_{tn}$ may still have negative eigenvalues.

My current perspective is that this largely inhibits us from transferring intuitions from the linear -> nonlinear case.

---

W2 - Literature placement.

The second paragraph of the introduction seems to suggest that the study of stable value function learning started in 2013 with the DQN paper and target networks. Really this has been a focus of study since the early 90s with a rich literature of proposed strategies.

One notable line of work is the gradient TD methods starting from  Antos et al. 2008 and Sutton et al. 2009. This seems particularly relevant because the regularizer in eqn 5 of this submission looks _quite_ similar to the TDC method from Sutton et al. 2009. Both are a correction of the standard semi-gradient TD update rule, both down-weighted by gamma, both dependent on the value function at the next state (i.e. $P \Phi w$), both using some lagged secondary weight vector. A few recent papers have proposed gradient TD methods as a strategy to remove the need for target networks in nonlinear learning.

Gradient TD methods also provide a solution to two of the open challenges in this work. GTD methods are provably convergent (even in the nonlinear setting) under broad conditions and even when TD is not convergent. Also TDC, specifically, is adaptive in the degree that it depends on its correction term. For functional regularization, $\kappa$ must be selected ahead of time through tuning. TDC uses an estimate of the TD-error to control the degree of regularization. When the value function is nearly accurate, we no longer need as much "correction" so the algorithm reverts to TD(0). When the value function is highly inaccurate, TDC more heavily leverages the regularization.

I certainly don't mean to suggest GTD methods subsume the proposed method here, as the perspectives are quite different and the resultant algorithm has some minor differences. But I do believe this submission is incomplete without a discussion on the relationship between these.

---

W3 - Stability.

This paper does a great job of showing that target networks can sometimes introduce novel unstable regions to the optimization process. However, it also has been shown that target networks can introduce novel _stable_ regions (e.g. off-policy TD(0) with target networks on Baird's counterexample converges). From my understanding, the proposed functional regularization does not change the stability region from standard TD(0) while continuing to pay the sample efficiency cost of target networks---and perhaps a worse sample efficiency cost according to Fig 1.

These theoretical and empirical findings in the linear case do an exceptional job of convincing me that target networks are not a valid solution for stabilizing TD learning. However, they do not convince me that functional regularization is a suitable replacement, in all cases the standard TD(0) seems strictly superior. The empirical findings in the nonlinear case are more motivating, but also much harder to understand and limited to a very "nice" environment: FourRooms. The empirical findings on Atari appear insignificant and are very likely due to the proposed method having additional degrees of freedom.

Ultimately, I strongly suspect something interesting is happening with functional regularization here and that it provides something beyond TD(0), but I'm not sure the linear FA results get to the heart of what that is. It doesn't appear to be stability, as suggested.

---

> ### Author Response · Authors · 2023-07-27
> **Answer to Rev GKoB**
>
> Hello,
>
> Thank you for your careful review, you made many comments showing a deep understanding of the paper and we wanted to address them directly here rather than in the general message.
>
> # Linear vs non-linear setting
>
> We fully acknowledge that a careful analysis in a non-linear setting would be valuable, however we have not managed to obtain conclusive results in that setting: quantifying regions where the method might converge or diverge becomes less interpretable as they depend on the optimization dynamics of the neural network.
> Your comment about the matrix $\Phi D \Phi_{tn}$ potentially having negative eigenvalues is correct, however when using deep networks, our loss may not even be convex in the first place so this is not the only problem we will have when analyzing convergence.
> We agree with your intuition that a reason why FR (and TN) are superior to TD in deep RL certainly has to do with the use of DNNs, however as far as we know, understanding the optimization dynamics and loss landscape of simple neural networks in a supervised learning setting is still an active area of research.
>
> # Stability
>
> Concerning the stability comment, we try to be very careful in the paper with respect to our theoretical claims. Our main claim is that TN, which can be seen as a regularization of TD, can make TD unstable. On the other hand, FR stays closer to the original algorithm in terms of convergence domain. Let us know if there are areas in the updated version which you think should be reformulated.
>
> However, **experimentally, we do see that TN and FR are indeed more stable than TD(0).** On Fig 3b and 5b we show that TD(0) typically finds very large Q values, far away from a reasonable range. And, as we increase the update period (up to a certain point), both FR and TN become better at approximating the true Q-values.
> This is corroborated by Fig 2 of “Deep Reinforcement Learning and the Deadly Triad” van Hasselt et al. which also shows that using a target reduces soft divergence.
>
> As you do, we do also believe that the explanation for the phenomenon cannot be entirely captured in the simple linear function approximation regime. We strongly suspect it is related to the fact that both the features and the linear weights on top of them are learned jointly, which might exacerbate divergences and that TN and FR regularization helps with that problem. While we are interested in investigating this question, we also believe this would be worth a standalone paper.
>
> Thank you again for your time and the effort you put into understanding our paper and helping us improve it!

---

> > ### Comment · Reviewer_GKoB · 2023-08-16
> > **Nonlinear theory**
> >
> > This comment is unrelated to whether the paper should be accepted in its current form---I am happy for theory in the nonlinear setting to be future work. I believe the current analysis yields insight, though I am uncertain of the utility of that insight. But in the spirit of TMLR, all insight is valuable and should be published!
> >
> > ---
> >
> > One clarification about my concerns with the linear vs. nonlinear theory. Here I am not simply asking for more, but rather pointing out that the theory in the linear setting may provide _no_ insight about the nonlinear setting. Here's an exaggerated metaphor:
> > > I provide some theoretical result showing that the materials used to build a basketball make the ball painful to kick. As a result, I conclude that a basketball is a poor choice for playing soccer. If I modify the material in a minor way, I can more comfortably kick the ball. My theoretical findings are correct and insightful in the kicking case, even though no one has ever considered using a basketball for soccer. Even after the material modification, people still prefer to use a soccer ball for kicking. Unfortunately, our findings were uniquely suited to kicking and provide no insights about dribbling on a wooden court, shooting to a basket, etc. The question is, are these insights useful?
> >
> > Typically, I am a strong proponent of starting theory in the linear case in order to inform decisions in the nonlinear case. But when the goal is exclusively to improve the nonlinear case, one has to motivate clearly why the investigation in the linear setting matters. If modifications make sense in the linear setting, but there is no clear analog in the nonlinear setting, it begs the question whether these modifications are useful. Take the GTD line of work. Despite the original theory being exclusive to the linear setting, the insight of using a two-timescale learning system in order to provide stability generalizes well to the nonlinear setting. While there was no immediately clear path for how to do the analysis in the nonlinear setting (though Hamid did publish a paper on this a year later), there was also no immediate red flag saying one _couldn't_ do the nonlinear setting. In the case of FR, the insight about the eigenvalues of the feature covariance matrix appears highly specific to the linear setting, a red flag that this insight would not hold in the nonlinear setting. The deviation of $\Phi$ vs $\Phi_{tn}$ when moving to the nonlinear setting is another red flag that this insight does not generalize.
> >
> > Some of these deviations could be investigated empirically. For example, take the $\Phi D \Phi_{tn}$ issue. Treating the neural network as frozen and producing these "feature" matrices from the penultimate layer of the neural network would allow us to empirically study the properties of this matrix. What do the eigenvalues look like as $\Phi_{tn}$ becomes more stale (i.e. as refresh rates are longer)? These experiments could be used to provide evidence that some part of the theoretical findings continue to hold even in the nonlinear setting (or possibly these experiments would show otherwise!). Possibly we could list each of the "red flags" that may inhibit theory in the nonlinear setting and empirically investigate whether these red flags actually hold.

---

### Author Response · Authors · 2023-07-25
**General Message (1/3)**

We would like to thank the reviewers for the time and effort they have spent on the paper, we think the points raised are pertinent. We will address here the main changes we made to the paper and we will answer individually and shortly points that have not been addressed in the general message.

**TLDR:** we have softened our claims on Atari and strengthened the evidences by running experiments over the whole Atari suite for TN and FR kappa = 1 and 2 network update periods for 7 seeds (sec 4.3). We added several related work citations and a connection to GTD method paragraph (sec 3.3)

# Paper update (in green in the pdf)

## Clarity and related work
----
> The paper would benefit from a richer discussion of placement in the literature. Specifically, there is no mention of gradient TD methods, and these share nontrivial similarities to the proposed approach.

> Several works were mentioned in the related work section, but I feel like the comparison to the closer 1 or 2 works should've been included somehow in the paper (in the experiments, or the theory). Otherwise the work feels a bit secluded.


We have added a discussion regarding gradient TD methods in Section 3.3 (addressing points raised by Rev GKoB and Rev pCv4).

----

> In Four Rooms and Atari games experiments, all returns are very wild since they all drop at the end, even worse than the initial return. Can you explain why?

> Also, the Atari experiments do not seem to have converged for all algorithms.

We have modified the captions of Fig 3 a) b) and Fig 5a) b) such that the ablations/robustness studies are not confused with learning curves which are provided in the appendix (Rev uvyZ)

----

> Eqn 3 really isn't the gradient of any loss function.

We have defined a semi-gradient operator $\tilde{\nabla}$ so as to differentiate it from the true gradient operator.

----

> At the top of page 3, it is stated that linear functions provide a poor approximation in complex environments.

Indeed, the text now reads: In complex environments, it is difficult to adequately design features to approximate $Q^\pi$ with a linear function.

----

> Align the Atari claims with the evidence---either tone down the claims to say that FR is at least as good as TN or ramp up the evidence to suggest FR is truly superior.

We have softened the Atari claims to “matches and sometimes outperforms TN” and strengthened the Atari results (see experiments)

----

> From my understanding, the proposed functional regularization does not change the stability region from standard TD(0)

We have reworded some uses of “stable” that could lead the reader to believe that FR is always more stable than TN. And change section 3 title from ``Functional Regularization for stable Reinforcement Learning`` to ``Functional Regularization as an Alternative to Target Networks``

----

---

> ### Author Response · Authors · 2023-07-25
> **General Message (2/3)**
>
> # Experiments
> ----
> > The paper claims superiority on Atari, but at best the experiments provide evidence of equivalency. Because the proposed method has an additional hyperparameter, and because no single setting of that hyperparameter performs consistently well.
>
> > The limitations mentioned by the authors themselves. I think the biggest issue for me is the extra parameter kappa, as it feels like some theoretical connection between kappa, gamma and T should exist.
>
> As requested by all reviewers, we compared FR without additional tuning (kappa=1) to TN on the complete Atari suite. We report that FR without tuning matches and sometimes outperforms TN. Our new results show that FR without tuning (kappa=1, T=1000)  obtains a mean normalized human score significantly higher than what is obtained by TN with its default hyperparameter, T=1000 (see Fig 4). We hope this can convince the reviewers that our method is simple to use, flexible, and does not need additional parameter tuning compared to TN on the Atari suite. See section 4.3 for all the details and statistical analysis of the results.
>
> ----
>
> > Perhaps the authors could observe the regularization effect in experiments, especially in general benchmark experiments through some ablation studies, to match the article's logic, rather than just directly observing the final experimental results. Because even if the experimental results improve, the logic in the previous text (FR improves TD learning through better regularization) cannot be verified.
> We have changed the legend of figures 3 and 5 for clarity but these figures actually show the final performance of FR, TN and TD(0) for various hyperparameters. As such they should be thought of as ablation/robustness studies and are not learning curves. We now provide all the learning curves for all Atari games in Fig 10 in the appendix (they are increasing as expected).
>
> On fig 3 and 5, TD(0), in yellow, is a single point since there is no regularization, while FR regularization can be controlled by changing the regularization weight (different shade of blue) and increasing the network update period (x axis). We believe these figures show that FR compared favorably to TD(0) for a wide range of hyperparameters (including the default kappa=1) and different environments (atari suite and four room).
>
> ----
>
> > If possible, please add to the supplementary the convergence plots of each Atari game (rewards of roll-out episodes) for the best set of parameters. It usually gives a better perspective on which algorithm is better.
>
> We include learning curves for every game in the appendix as requested by Rev pCv4
>
> ----
>
> > Regarding computation overhead, how is FR compared to using TN?
>
> As requested by Rev nz5F09, In figure 6, we have analyzed the additional computation overhead of FR compared to TN. On the Atari games we have not observed a large difference between TN and FR run times since the interactions with the simulator dominate the runtime compared to the backpropagation time.

---

> > ### Author Response · Authors · 2023-07-25
> > **General Message (3/3)**
> >
> > # Theory
> > ----
> > Concerning our theoretical setting: we have two main theoretical results in the linear function approximation regime, one for TN and one for FR. While the setting is simple, the analysis of these algorithms is not as simple as they seem as both algorithms both have an inner loop and outer loop component and their interaction needs to be understood carefully. Furthermore while the two algorithms appear similar, their analysis reveals some profound differences in terms of the regions where each algorithm is stable.
> >
> > ----
> >
> > > There is an unfortunate limitation that theoretical results are exclusive to the linear setting, while target networks are almost exclusively used in nonlinear settings. This deviation nullifies one of the major motivations of the proposed algorithm.
> >
> > We believe this setting is appropriate for TN as we show a negative result, i.e the fact that TN can introduce regions of divergence that may not be present while simply using TD. As these phenomena only appear in the presence of off-policy learning, function approximation and the use of bootstrapping, this is the simplest and thus strongest setting for which we can show that TN can introduce new divergence regions.
> > Importantly, we believe this result is at odds with beliefs within the reinforcement community, thus very valuable. For instance Zhang et al, ICML 2021 (https://arxiv.org/abs/2101.08862) showed that a modified version of TN can converge. More recently Fellows et al, ICML 2023 (https://arxiv.org/abs/2302.12537) claimed that TN stabilizes TD, even in the non-linear regime. However this last paper was at odds with our theoretical and empirical results and after discussion with the authors, they have since acknowledged an important error and retracted their paper.
> >
> > ----
> >
> > > While the theoretical part of the paper is limited to a very simple setting, which is linear function approximation with finite state and action spaces, and no exploration effect is considered, the effectiveness of the proposed method should be evaluated carefully using experimental methods.
> >
> > However, as pointed out by Rev GKoB and uvyZ, we agree that the analysis of FR would benefit from being extended to the stochastic and potentially nonlinear function approximation, however the proofs are not as straightforward as the TD(0) case as the iteration matrix is now more complex. Despite this, we believe that we see meaningful differences between FR and TN in that setting already as exemplified on Fig 1 and that because of our other theoretical and experimental results our paper already makes valuable contributions to the community.

---

### Comment · Action_Editors · 2023-08-27
**Regarding Farahmand et al. (2009)**

(I am writing not as an action editor, but as an author whose research has been cited in this work.)

The paper mentions Farahmand et al. (2009) and states that "Farahmand et al. (2009) is perhaps the closest work to our own, they regularize the L2 norm of the Q-value estimates, i.e., penalizing $κ \||Q_w||^2$. However, penalizing the magnitude of the Q-values would prevent the algorithm from converging to $Q^*$ if $κ$ does not tend towards 0."

I am glad that you cited this work, even though it is from years ago -- relatively uncommon these days.

I'd like to make some comments about the interpretation of the regularizer in Farahmand et al. (2009) work:
1) That work started by allowing any choice of regularizer $Pen(Q)$ and not necessarily the $L_2$ norm of $Q$. Some mentioned examples were Sobolev-space norm (which has a smoothness interpretation) and the RKHS norm.
2) The derived formulaee (e.g., Equation 3 in the paper) and the theoretical results are for the choice of function space being an RKHS and the penalty being its RKHS norm.
3) An RKHS or Sobolev space norms are not generally the magnitude ($L_2$ or similar) of $Q$. For example, a Sobolev norm measures the smoothness of a function, in addition to its magnitude.
4) As noted by the authors, the regularizer coefficient in that work should go to zero as the number of data points goes to infinitey. The optimal rate is provided just after Theorem 2.

I'd also mention that this has been developed and expanded in my PhD work (Chapter 5): https://sologen.net/papers/RegularizationInReinforcementLearning(PhD-Dissertation-Farahmand).pdf

---

### Decision · Action_Editors · 2023-08-27

**Recommendation:** Accept as is

**Comment:**

The paper suggests modification to a widely-used algorithm, supported by empirical and intuitive justifications that are likely to capture community interest. The raised concerns by the reviewers have been addressed in the rebuttal, and the paper contributes insights toward enhancing stable reinforcement learning. The revisions adequately address the initial concerns expressed in the reviews.

I encourage the authors to address any minor issues before submitting their camera ready version.

**Audience:**

This paper is of interest to the broad subset of the reinforcement learning community, and as such, TMLR readers.

**Claims And Evidence:**

All reviewers believe that there are enough evidence for the claims of the paper.